# Resistance to CDK7 inhibitors directed by acquired mutation of a conserved residue in cancer cells

Chun-Fui Lai [ID][1], Victoria I Cushing[2], Ellen Olden [ID][1], Charlotte L Bevan [ID][1], R Charles Coombes [ID][1], Basil J Greber [ID][2,✉], Laki Buluwela [ID][1,✉] & Simak Ali [ID][1,✉]

## Abstract

CDK7 has emerged as a cancer target because of its pivotal roles in cell cycle progression and transcription. Several CDK7 inhibitors (CDK7i) are now in clinical evaluation. Identifying patients most likely to respond to treatment and early detection of tumour evolution towards resistance are necessary for optimal implementation of cancer therapies. Continuous culturing of prostate cancer cells with Samuraciclib, a non-covalent ATP-competitive CDK7i, led to outgrowth of resistant cells. These were characterised by the acquisition of a single base change in the CDK7 gene, Asp97 to Asn (D97N). Mutant cells were resistant to other non-covalent CDK7i but remained sensitive to covalent CDK7i. Cryo-EM structure and kinase ligand affinity determinations revealed reduced affinity of the CDK7-D97N mutant for non-covalent CDK7i. Remarkably, Asp97 is absolutely conserved in human CDKs, inferring its importance for the activities of all CDKs. Consistent with this, mutation of the homologous residue in CDK12 (D819N) or CDK4 (D99N) promoted resistance to drugs that inhibit these CDKs. Our findings reveal a general mechanism for acquired resistance with obvious implications for patients treated with CDK inhibitors.

**Keywords** Cancer; Cyclin-dependent kinases; Drug resistance; Tumour evolution
**Subject Categories** Cancer; Cell Cycle; Pharmacology & Drug Discovery

## Introduction

Cyclin-dependent serine/threonine kinases (CDKs) play critical roles in the regulation of cell cycle and transcription processes (Malumbres, 2014). Thus, the actions of CDK4, CDK6, CDK2 and CDK1 are necessary for appropriate transition through the growth, DNA replication and mitotic stages of the cell cycle in metazoans (Morgan, 1997). Dysregulation of processes controlling cell cycle progression involving cell cycle CDKs is common in cancer, frequently through elevated and/or inappropriate activation of CDKs and/or their cyclins (Matthews et al, 2022; Suski et al, 2021).

Aberrations in cell cycle checkpoint controls resulting in loss of expression/activity of genes that restrict CDK activity are also regularly seen in cancer. This understanding has fuelled the development of small-molecule inhibitors. Most success has been achieved with CDK4/6 inhibitors (CDK4/6i) for oestrogen receptor (ER)-positive breast cancer. CDK4/6i combined with endocrine therapies that inhibit ER show a survival benefit in patients with advanced ER+ breast cancer (Morrison et al, 2024). Indeed, CDK4/6i have become standard-of-care treatments in advanced ER+ breast cancer patients and have progressed to the adjuvant setting in some patients with early-stage disease.

CDK7, CDK9, CDK12, CDK13, CDK8 and CDK19 are required for RNA polymerase II (PolII) mediated transcription initiation, elongation, termination, as well as RNA splicing, and for enhancer functions (Fant and Taatjes, 2019; Hsin and Manley, 2012). A common substrate for transcriptional CDKs is the C-terminal domain (CTD) of the largest PolII subunit, comprised of 52 repeats of a seven amino acid sequence motif, having the consensus sequence $Y_1S_2P_3T_4S_5P_6S_7$. The sequential action of CDK7, CDK9, CDK12 and CDK13 on different serine residues at the PolII CTD drives progression through the "PolII transcription cycle": transcription initiation, elongation, termination and recycling of unphosphorylated PolII (Parua and Fisher, 2020). Dysregulation of gene expression programmes is common in most cancers and mutations in transcription factor (TF) genes is often observed (Kandoth et al, 2013; Lawrence et al, 2014). Examples include the *MYC* oncogene, which is the most frequently amplified gene across cancer types (Beroukhim et al, 2010; Lin et al, 2012). ER and the related androgen receptor are critical transcriptional drivers in breast and prostate cancer, respectively, as demonstrated by the clinical utility of drugs that prevent or inhibit their activation (Ali et al, 2011; Rebello et al, 2021). Mutation, rearrangement and altered expression of genes encoding chromatin remodelling and modification enzymes (e.g. EZH2, CBP/p300, BRD4, ARID1A) is further confirmation of aberrant transcriptional processes in cancer (Kandoth et al, 2013; Lawrence et al, 2014). Transcriptional dependencies in many cancers, together with identified roles in regulation of TF activities and epigenetic regulation, as well as DNA damage responses, have highlighted transcriptional CDKs as cancer targets (Chou et al, 2020; Parua and Fisher, 2020).

CDK7, because of its dual roles in regulating the activities of both the cell cycle and transcriptional CDKs (Fisher, 2019), has

[1]Department of Surgery & Cancer, Imperial, London, UK. [2]Institute of Cancer Research, Chester Beatty Laboratories, London, UK. ✉E-mail: basil.greber@icr.ac.uk; l.buluwela@imperial.ac.uk; simak.ali@imperial.ac.uk

received particular attention for drug development. CDK7, in a trimeric complex with cyclin H and the accessory protein MAT1, forms the CDK activating kinase (CAK). Regulation of the cell cycle by CAK is mediated through its phosphorylation of CDK1, CDK2, CDK4 and CDK6 in the T-loop to promote kinase activity and facilitate/stabilise complex formation with cognate cyclins (Schachter and Fisher, 2013; Schachter et al, 2013). CAK, as part of the general transcription factor TFIIH, phosphorylates serine-5 and serine-7 in the PolII CTD heptad repeat, a step necessary for transcription initiation by PolII. By also phosphorylating CDK9, CDK12 and CDK13, which themselves regulate transcription elongation, termination and splicing, CDK7 impacts all stages of transcription (Larochelle et al, 2012; Rimel et al, 2020).

Several CDK7i, both high-affinity ATP-competitive and covalent CDK7i have now been described (Sava et al, 2020). Pre-clinical studies using these drugs reveal broad cancer potential for CDK7i (Ali et al, 2009; Chipumuro et al, 2014; Christensen et al, 2014; Kwiatkowski et al, 2014; Patel et al, 2018; Wang et al, 2015). CDK7 inhibition by these drugs reduces PolII phosphorylation and represses transcription. Reduced T-loop phosphorylation of cell cycle CDKs has also been demonstrated. Several CDK7i have progressed to clinical trials, and we recently reported that Samuraciclib (ICEC0942, CT7001) (Patel et al, 2018) shows clinical efficacy in ER+ breast cancer patients who had progressed on prior CDK4/6i (Coombes et al, 2023). However, markers of response are urgently required for developing the most appropriate clinical utilisation of these new cancer drugs.

Our previous work demonstrated Samuraciclib sensitivity of prostate cancer cells in vitro and in vivo (Constantin et al, 2023). By continuous culturing of prostate cancer cells to develop resistance to Samuraciclib, we have identified a new mechanism of resistance to CDK7i. Samuraciclib-resistant prostate cancer cells were characterised by the acquisition of a mutation that alters a single amino acid, Asp97. This change in CDK7 is sufficient for resistance to multiple non-covalent, ATP-competitive CDK7i. Since this aspartate is absolutely conserved in all human CDKs, our findings also have broader implications. Indeed, mutating the analogous residue in CDK4, CDK6 or CDK12 causes resistance to selective inhibitors of these CDKs. Together, our findings reveal a common mechanism for the acquisition of resistance that could have important ramifications for the clinical use of CDK inhibitors in cancer patients.

# Results

## Generation of prostate cancer cell lines with acquired resistance to Samuraciclib

To model acquired resistance in cancer, we cultured the 22Rv1 prostate cancer cell line that was originally generated from a human prostatic xenograft as a model of castrate-resistant prostate cancer (Sramkoski et al, 1999), in medium containing a moderately high concentration of Samuraciclib, five times higher than the half-maximal inhibitory concentration ($IC_{50}$) concentration (Fig. EV1A), with medium changes every 3 days. Cell numbers remained low, but growth was re-established, allowing cell expansion within 4 months after initiation of drug treatment (Fig. 1A). Growth inhibition assays showed that Samuraciclib-resistant 22Rv1

(22Rv1-SamR) cells were 16-fold less sensitive to Samuraciclib than 22Rv1 cells, with $IC_{50}$ values of 1719 and 107 nM, respectively (Figs. 1B,C and EV1A). 22Rv1-SamR cells were cross-resistant to other ATP-competitive CDK7i, with 45- and 1378-fold reduction in sensitivity for LDC4297 (Hutterer et al, 2015) and SY-5609 (Marineau et al, 2022), respectively. Interestingly, 22Rv1-SamR cells remained sensitive to the covalent CDK7i, SY-1365, YKL-5-124 (Olson et al, 2019) and THZ1 (Kwiatkowski et al, 2014). Phosphorylation of the PolII CTD at Ser5 is the best characterised site phosphorylated by CDK7, an event that is necessary for transcription initiation (Fisher, 2019). Ser5 phosphorylation was reduced by samuraciclib and by SY1365 in 22Rv1 cells (Figs. 1D and EV1B). In 22Rv1-SamR cells, Ser5 phosphorylation was not reduced by Samuraciclib but was significantly lower for cells treated with SY1365. Consistent with its role as the CAK activating kinase CAK, CDK2 phosphorylation at Thr160 (Merrick et al, 2008) was also reduced by Samuraciclib and by SY1365 in 22Rv1 cells. While SY1365 treatment reduced CDK2 phosphorylation in 22Rv1 cells, its phosphorylation by samuraciclib was only evident at the high dose of 1000 nM. These results are consistent with resistance to samuraciclib but not to the covalent CDK7i, identified in the above growth assays. Note that CDK7 protein levels were not much altered, although it appeared to migrate further in lysates from 22Rv1-SamR than in lysates from 22Rv1 cells.

## Resistance to Samuraciclib is driven by a single amino acid change in CDK7

We carried out RNA-seq to identify gene expression changes that accompany Samuraciclib resistance in 22Rv1 cells. Expression levels of MAT1 (MNAT1) and cyclin H (CCNH) were unaltered, although CDK7 expression was modestly reduced in 22Rv1-SamR cells (Fig. EV1C). Interestingly, a single nucleotide difference in exon 5 (c.289G>A) was apparent in 22Rv1-SamR cells, with the variant form being present in a 1:1 ratio with the wild-type (WT) CDK7 sequence, suggestive of a single allele alteration (Fig. EV1D,E). Sanger sequencing of genomic DNA and cDNA confirmed the presence and expression of the c.289G>A variant in 22Rv1-SamR cells (Fig. 1E). This base change would result in a single amino acid change, p.Asp97Asn (D97N).

To determine if the CDK7-D97N mutation is sufficient for resistance to CDK7i, we used CRISPR-Cas9 to introduce this mutation in prostate cancer cells (22Rv1, LNCaP) and a breast cancer (MCF7) cell line. Following ribonucleoprotein (RNP) complex delivery of a CRISPR targeting CDK7 exon 5 and donor template incorporating a GAT (Asp) to AAT (Asn) change at codon 97 (Figs. 2A and EV2A,B), culturing MCF7 cells with samuraciclib for 3 weeks was sufficient to enrich for the CDK7-D97N mutation (Fig. 2B). In growth assays, MCF7-CDK7-D97N cells were less sensitive than WT cells to samuraciclib (45.4-fold), LDC4297 (38.8-fold) and SY-5609 (1105-fold) (Fig. 2C,D). As observed in 22Rv1-SamR cells, sensitivity to covalent CDK7i was not reduced in CDK7-D97N cells. Blunting of the growth inhibition by samuraciclib was confirmed by the smaller impact of samuraciclib on inhibition of CDK7 target protein PolII and CDK2 phosphorylation (Fig. 2E). By contrast, inhibition of PolII and CDK2 phosphorylation by the covalent CDK7 SY1365 was maintained.

Using the same CRISPR approach as that used for MCF7 cells, LNCaP and 2Rv1 cells similarly enriched for mutant cells

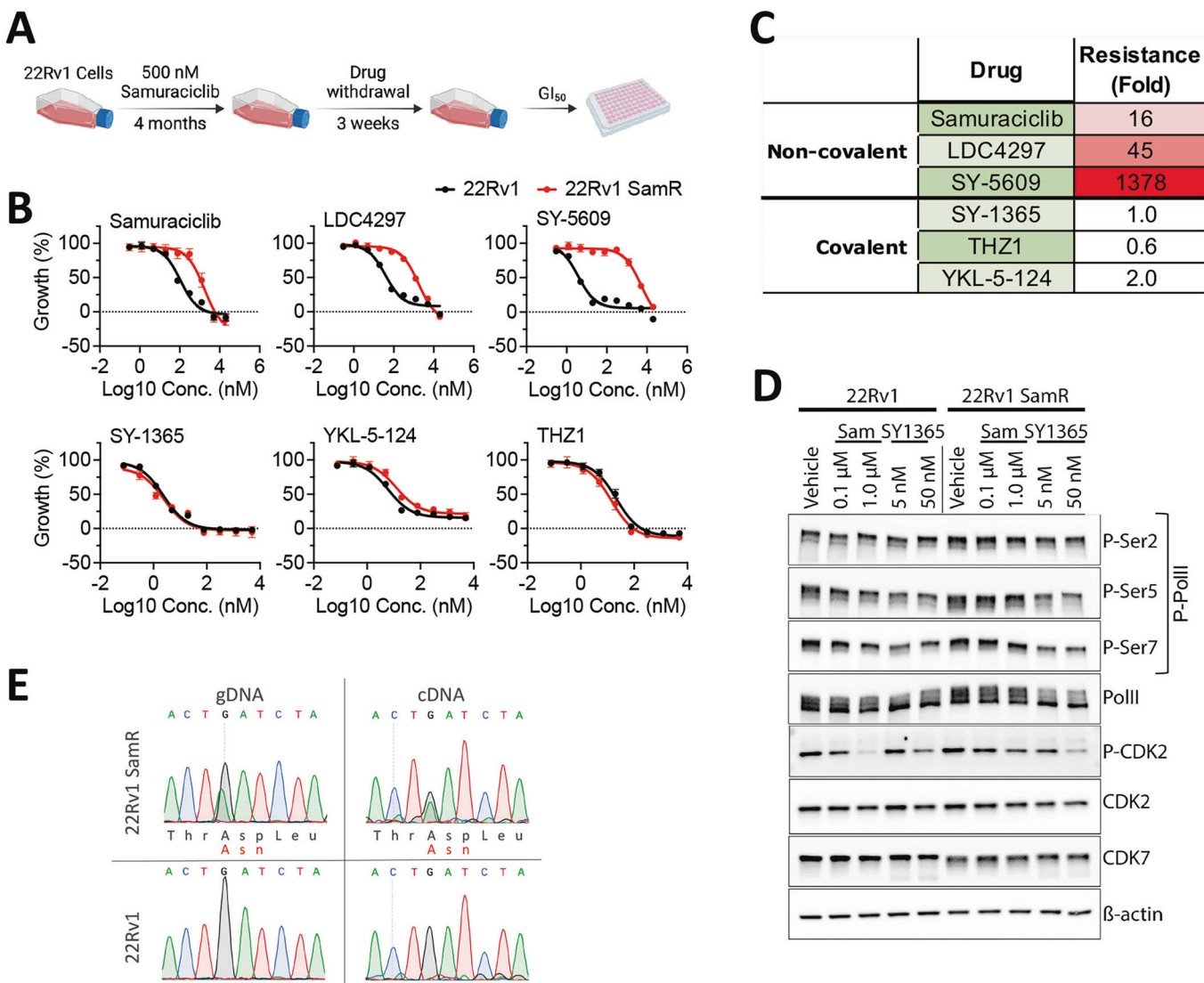

**Figure 1. Development of resistance to CDK7 inhibitors by acquisition of a CDK7-D97N.**

(A) Summary of the scheme followed for the generation of resistance to Samuraciclib (Sam) in 22Rv1 cells. (B) Mean growth relative to vehicle (DMSO) following addition of increasing concentrations of the named drugs for 4 days (Samuraciclib $n = 3$ independent experiments; other drugs $n = 2$; error bars = SEM). (C) Shown are the fold increases in IC$_{50}$ values of 22Rv1-SamR cells relative to those for 22Rv1. (D) Cells were treated with Sam or SY1365 at the concentrations indicated. Protein lysates prepared 24 h following drug addition were used for immunoblotting using antibodies for the indicated proteins and phosphorylation marks. An equal volume of DMSO was added to the vehicle controls. (E) Sequencing chromatograms for PCR products using genomic DNA (gDNA) or cDNA in the region of the CDK7 gene encoding aspartate 97. Source data are available online for this figure.

(Fig. EV2C), with the mutant cells being substantially less sensitive to the non-covalent CDK7i while remaining sensitive to the covalent CDK7i, SY-1365 and YKL-5-124 (Fig. EV2D). The CRISPR knock-in data for multiple cell lines show that the samuraciclib resistance acquired by 22Rv1 cells is mediated by the CDK7-D97N mutation and is unlikely to involve alterations in other genes. Finally, we observed that the growth rates of 22Rv1 and 22Rv1 SamR cells in the absence of CDK7i are extremely similar over time (Fig. EV2E), with calculated doubling times of 49.4 (95% CI = 45.3–54.1) and 45.4 h (95% CI = 42.2–49.0), respectively. We also assessed the growth of a 22Rv1 CRISPR clone homozygous for the D97N mutation (Fig. EV2F). This clone

had a doubling time of 48.9 h (95% CI = 45.1–53.0 h). By contrast, 22Rv1-SamR and the homozygous CDK7-D97N mutant clone were considerably less growth inhibited by 1.35 μM samuraciclib than 22Rv1 cells. All three cell lines were potently growth inhibited by SY1365. These results suggest that the CDK7 D97N mutation does not adversely affect cell fitness, at least in vitro.

### The CDK7-D97N mutation reduces the binding affinity of ATP-competitive CDK7i

To directly assess the effect of the D97N mutation on CDK7 activity, we turned to the NanoBRET method in which CDK7-WT and CDK7-97N were fused to NanoLuc. Energy transfer between

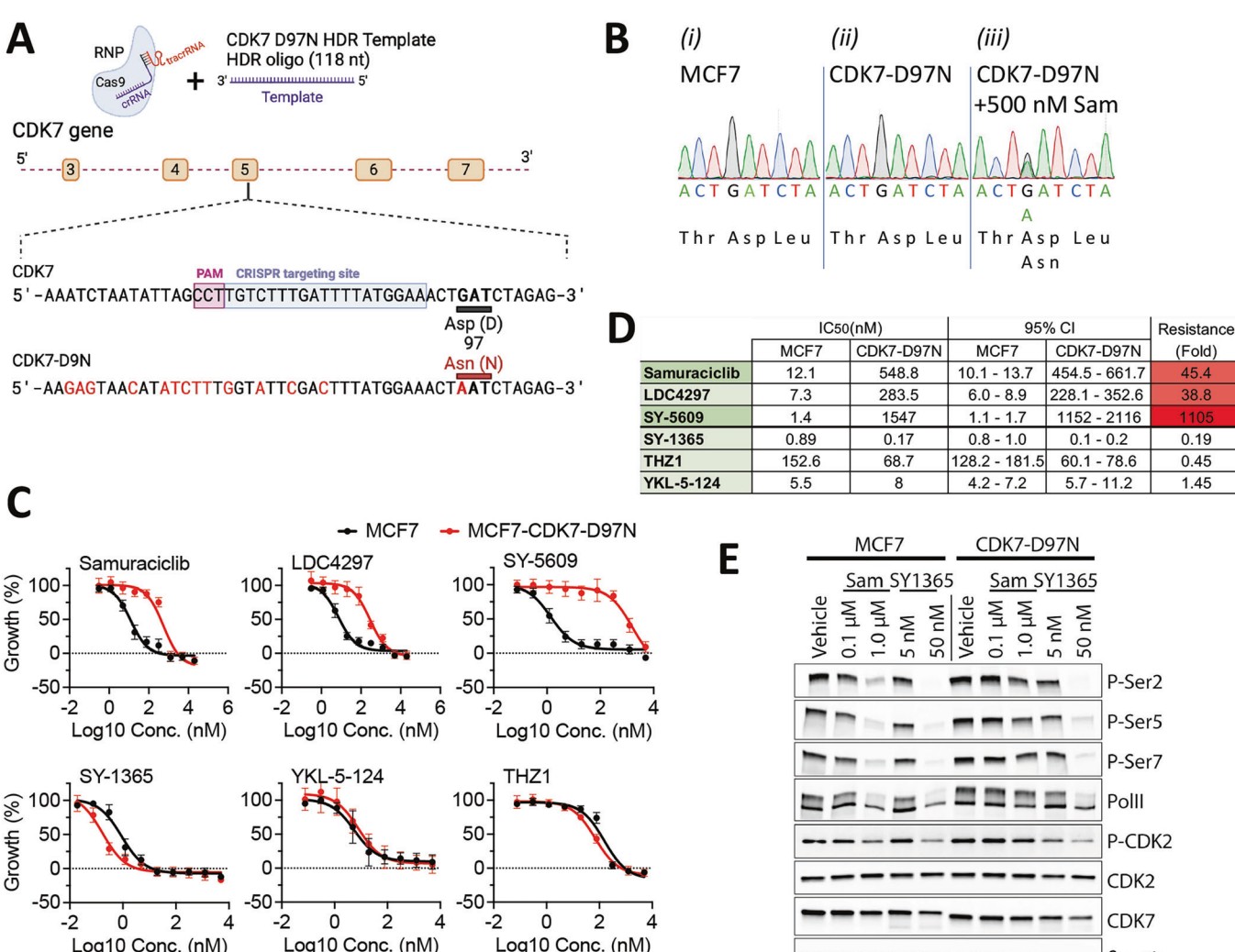

**Figure 2. Generation of the D97N mutation in MCF7 cells using CRISPR/Cas9 mutagenesis.**

(A) Schematic summarising the strategy for generating the D97N mutation. Also shown are the CRISPR sequences and partial sequence of the donor template, designed to incorporate non-coding changes (in red) in exon 5 to facilitate screening for successful knock-in. (B) Sequencing chromatograms of genomic DNA from MCF7-WT (i), MCF7 following nucleofection as in (A) cultured without Sam (ii) or after 3 weeks culturing in the presence of 500 nM Sam (iii). (C) Growth response to increasing concentrations of the indicated drugs using MCF7 or MCF7-CDK7-D97N cells that had been selected as in (B) (iii) demonstrates reduced sensitivity of the mutant cells to non-covalent CDK7i but maintained sensitivity to covalent CDK7i. Plots show means and SEM for $n = 4$ independent experiments, except for LDC4297 and THZ1 where $n = 3$. (D) IC$_{50}$ values have been calculated from the plots in (C). (E) Immunoblotting was performed using lysates prepared from cells treated with Sam or SY-1365 for 24 h. An equal volume of DMSO was added to the vehicle controls. Source data are available online for this figure.

the NanoLuc fusion protein and a cell-permeable fluorescent energy transfer probe facilitates quantitative determination of target engagement in live cells (Wells et al, 2020). Assays were conducted according to this previous report, using the optimal CDK7 probe, K-10, at a concentration of 0.5 μM. Tracer displacement was greatly reduced for Samuraciclib and SY-5609 for the mutant as compared with WT CDK7 (Fig. 3A–D), in line with the growth resistance. CDK7-D99N EC$_{50}$ values were 178- and >7000-fold lower for Samuraciclib and SY-5609, respectively, than CDK7-WT. By contrast, there was no significant difference in EC$_{50}$ values for mutant and WT CDK7 in the case of the covalent CDK7i, THZ1 and SY-1365. These live cell CDK7 binding assays indicate that the D97N mutation reduces affinity for ATP-competitive CDK7i, while the covalent tethering means that there is no change

in binding, and subsequent displacement, of the probe in the case of covalent CDK7i.

To directly assess the enzymatic activity of CDK7-D97N in the presence of ATP, we assessed the activity of purified CAK-WT and CAK-D97N on T-loop phosphorylation of recombinant substrate CDK2. Quantification of immunoblotting showed that CAK-D97N is enzymatically active but that its activity is significantly lower than that of CAK-WT (Fig. 3E,F).

## Structure determination of ligand-bound CDK7-D97N by cryo-EM

To investigate the molecular basis of resistance of the CDK7-D97N mutant protein, we determined the cryo-EM structures of CDK7-

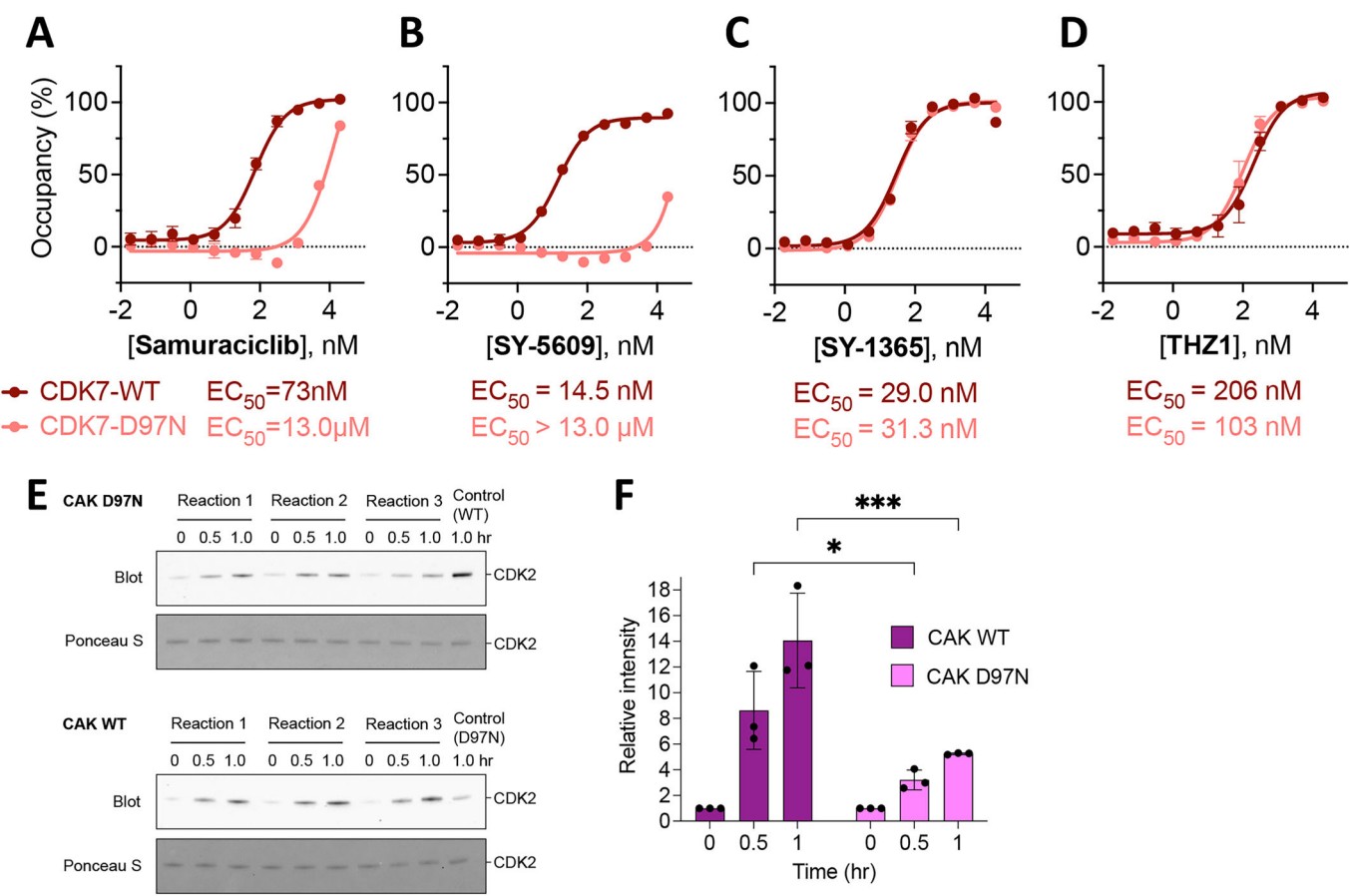

**Figure 3. Live cell CDK7i binding assays demonstrate reduced binding of non-covalent CDK7i to D97N-mutant CDK7.**

(A–D) HEK293 cells were transfected with Nano-luc CDK7 fusion constructs to investigate binding of different CDK7i, using the NanoBRET Target Engagement assay for fluorescent tracer (K-10) displacement ($n = 3$ independent experiments; error bars = SEM). Occupancy (%) represents the CDK7 fractional occupancy of CDK7 as a measure of tracer displacement with increasing concentration of CDK7i. (E) Western blot detection of T-loop phosphorylated CDK2 from three kinase reactions using recombinant CAK D97N (top) or CAK WT (bottom) and recombinant CDK2 as the substrate. Ponceau S staining was used to provide a loading control. Each membrane contains one control lane where the product of a 1-h reaction using the other CDK7 variant was detected to confirm successful blotting and detection and facilitate comparison of the results. (F) Quantification of the Western blot band intensities shown in (E), presented as the mean ± SD of $N = 3$ technical replicates. Significance levels were assessed using two-way ANOVA with correction for multiple comparisons (*$P < 0.05$; ***$P < 0.001$). Source data are available online for this figure.

D97N in the fully assembled human CAK, bound to the nucleotide analogue ATPγS, the ATP-competitive inhibitor Samuraciclib, and the covalent inhibitor THZ1 (Fig. 4A–C). The quality of our cryo-EM maps at 2.3–2.6 Å resolution (Fig. EV3A–I) allowed identification and modelling of the ligands in all cases (Figs. 4D–F and EV3J–L). Our structure of the complex bound to ATPγS shows that the overall architecture of the CDK7 active site and the mode of binding to adenine-nucleotides is preserved between the mutant and WT proteins (Fig. 4A,G) (Cushing et al, 2024). The differences between the cryo-EM densities of the nucleotide-bound mutant and WT structures are mostly restricted to the phosphates of the nucleotide analogue; however, the density in this area is less well defined than for the solidly bound adenine moiety, indicating flexibility. Differences of this magnitude have been observed for the WT enzyme before (Cushing et al, 2024; Greber et al, 2020) and are unlikely to reflect functionally relevant changes.

The structure of Samuraciclib bound to the mutant complex shows that the inhibitor, after incubation with the kinase at 50 µM concentration, predominantly binds in the same overall pose as

observed previously in the context of WT CDK7 (Fig. 4B,H) (Cushing et al, 2024). However, the density for the Samuraciclib hydroxypiperidine group is weaker than in prior cryo-EM reconstructions, and weak density for a second conformation of this chemical group is observed, indicating an increased level of structural heterogeneity in the context of the mutant protein (Fig. 4E). This observation suggests that the affinity of the inhibitor to CDK7-D97N might be reduced, as the contribution of the hydroxypiperidine group to inhibitor binding could be reduced or lost.

The structure of CAK (CDK7-D97N) modified by the covalent inhibitor THZ1 shows the inhibitor primarily bound in the same pose as observed in prior studies (Fig. 4C,I) (Cushing et al, 2024; Greber et al, 2020), in agreement with the observation that CDK7-D97N is still susceptible to this compound (Figs. 1B,C and 3). The head group is buried in the active site of the kinase, while the rod-like extension that carries the reactive group protrudes outside the active site cavity towards C312, which is covalently modified. In our cryo-EM map, we observe weak density that appears to originate

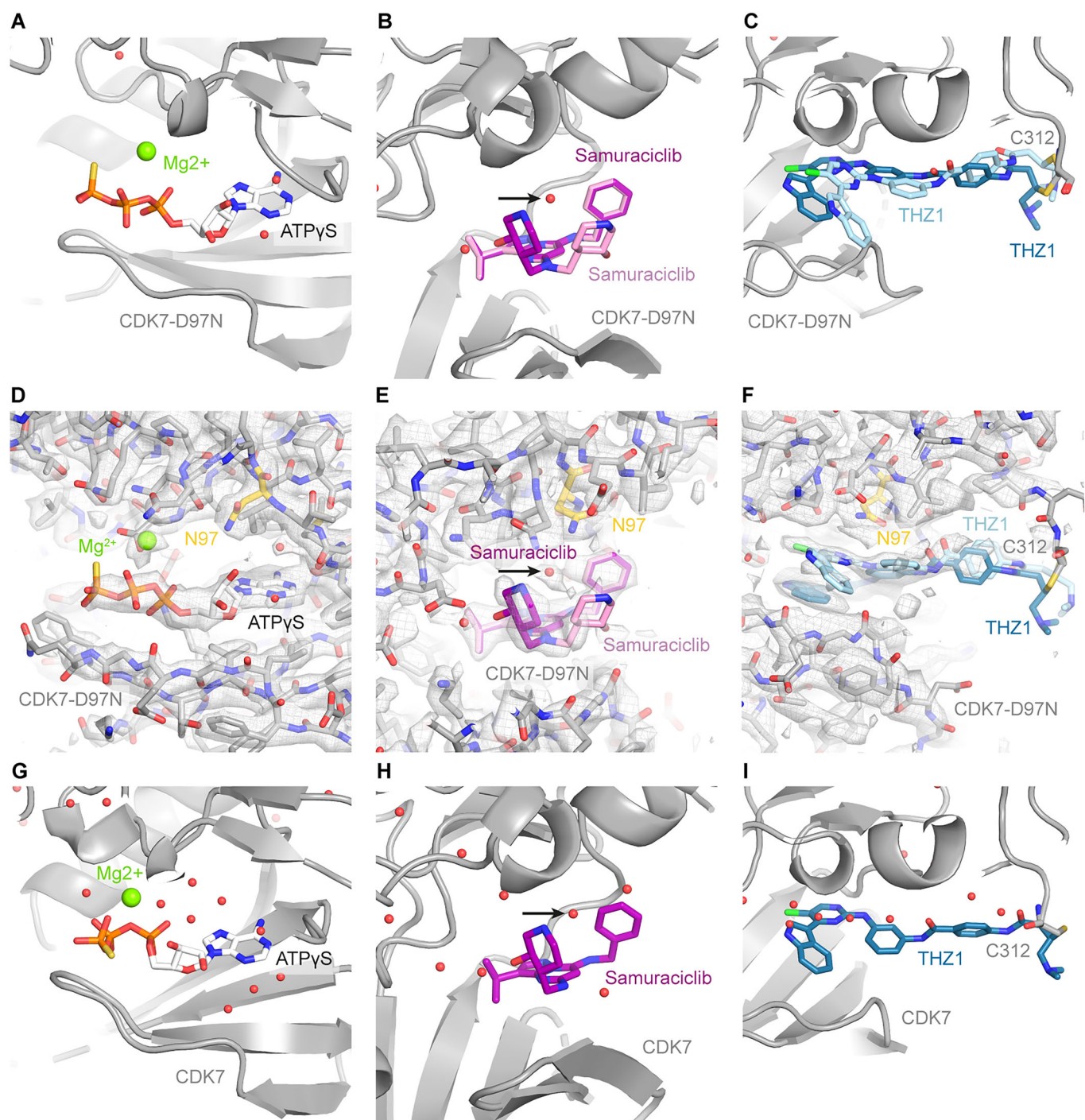

**Figure 4. Cryo-EM structures of CAK (CDK7-D97N) in complex with bound ligands and comparison to structures of wild-type CAK.**

(**A**) Structure of CDK7-D97N in complex with ATPγS. CDK7 is shown in grey, ATPγS in white, a magnesium ion in green, and water molecules in red. (**B**) Structure of CDK7-D97N in complex with Samuraciclib. Two conformations of the inhibitor are shown in purple and pink. A water molecule discussed in the text is indicated with a black arrow. (**C**) CDK7-D97N modified by THZ1. Two conformations of THZ1 are shown in teal and cyan. (**D–F**) As in (**A–C**), but shown with the cryo-EM density as a semi-transparent mesh and surface. Protein residues are shown as sticks, and the mutated N97 side chain is shown in yellow. (**G–I**) As (**A–C**), but for the wild-type CDK7 complexes (PDB IDs 8ORM, 8P6V, 8P6Y) (Cushing et al, 2024).

from an alternative conformation of the inhibitor (Figs. 4F and EV3l). In our prior studies with WT CDK7 (Cushing et al, 2024), this density was weaker and more fragmented and consequently interpreted as ordered water molecules (Fig. 4I). However, as it has not been possible to computationally separate cryo-EM particle sub-populations for one and the other inhibitor conformation, it is difficult to assess whether the alternative conformation from which the stronger appearance of this density in the CDK7-D97N mutant originates is unique to the mutant protein or just somewhat more strongly populated in the context of the D97N mutant than in WT CDK7.

Overall, our structural analysis indicates that the CDK7-D97N mutation does not induce major alterations of the CDK7 active site, consistent with the observation that the mutant enzyme is still active. While the loss of the negative charge on the side chain in the D97N mutation could conceivably directly impact interactions with certain inhibitors such as SY-5609, which forms a salt bridge to this residue (Mukherjee et al, 2024), rationalising its effects on other inhibitors is less straightforward. The hydroxypiperidine nitrogen in Samuraciclib forms a hydrogen bond to a water molecule that in turn hydrogen bonds to CDK7-D97 (Cushing et al, 2024). However, this water molecule is still present in the CDK7-D97N mutant structure (Fig. 4E). Similarly, LDC4297, which is also affected by the D97N mutation (Fig. 1B), does not appear to form any polar interactions with CDK7-D97 (Cushing et al, 2024). Overall, this suggests that subtle structural differences, which may be propagated by hydrogen bonding networks via water molecules, may contribute to the observed functional differences in ligand-binding properties that lead to cellular resistance to certain classes of inhibitors.

## Asp97 in CDK7 is conserved in human CDKs and its mutation causes resistance to ATP-competitive inhibitors of other CDKs

Amino acid sequence alignment reveals that CDK7-D97, as well as leucine at position 98, are invariant in all human CDKs (Wood and Endicott, 2018), 2 of just 29 out of 329 amino acids in human CDK7 that are identical in all human CDKs (Fig. 5A). In the CAK structure, D97 sits on the C-terminal side of the hinge region that connects the N- and C-terminal kinase lobes (highlighted in Fig. 4D–F), with its position conserved in the hinge region in other CDKs (Wood and Endicott, 2018). The complete preservation of this aspartate residue implies a conserved role, raising the possibility that its mutation to asparagine may impact response to ATP-competitive drugs as revealed herein for CDK7. We undertook experiments with two clinically relevant CDKs to directly consider this possibility.

CDK12 regulates transcription elongation and termination by CTD phosphorylation that involves phosphorylation at Ser2 but it is also implicated in pre-mRNA splicing. While the detection of inactivating CDK12 mutations in cancer indicates a tumour suppressor role, loss of CDK12 sensitises cancer cells to DNA damaging agents, at least in part due to the apparent CDK12 dependency for expression of larger genes with considerable numbers of exons, particularly genes involved in DNA repair, including *BRCA1*, *ATR*, *FANCI* and *FANCD2* (Liu et al, 2021; Pilarova et al, 2020). Thus, loss of CDK12 was identified as sensitising cancer cells to PARP inhibitors (Bajrami et al, 2014).

Moreover, increased genomic instability and intron retention in tumours with CDK12 loss have been associated with greater immunogenicity and potential susceptibility to immunotherapy agents (Antonarakis et al, 2020; Wu et al, 2018). These, as well as possible oncogenic actions through hyperactivation of CDK12 (Houles et al, 2022), have galvanised the development of CDK12i.

The importance of CDK12-D819 (equivalent to CDK7-D97) in the activities of CDK12i is indicated by the fact that D819 forms hydrogen bonds with CDK12i in co-crystallisation studies (Schmitz et al, 2024; Yan et al, 2023). We determined if replacement of D819 by Asn promotes resistance to CDK12i, using CRISPR-Cas9 to introduce the D819N mutation in the CDK12 gene. Successful knock-in was achieved, as revealed by PCR of genomic DNA using primers specific for incorporation of the mutant donor template (Fig. EV4A,B), albeit at levels that were too low for detection by Sanger sequencing of gDNA amplified using PCR primers flanking the exon encoding D819 (Fig. 5B (ii)). Culturing CRISPR-targeted MCF7 cells for 4 weeks with (R)-CR8, a molecular glue CDK12/CDK13/cyclin K degrader that binds in the CDK12/13 active site (Slabicki et al, 2020), enriched for D819N mutant cells (Fig. 5B (iii)). MCF7-D819N cells, following withdrawal of the drug for 1 month, showed a 12.6-fold reduction in sensitivity to (R)-CR8 in growth assays (Fig. 5C,D). In WT MCF7 cells, (R)-CR8 reduced PolII Ser2 phosphorylation, as well as levels of cyclin K and CDK12. By contrast, PolII phosphorylation was not inhibited and CCNK and CDK12 levels were not reduced by (R)-CR8 in MCF7-CDK12-D819N cells. There was no significant difference between the cell lines in growth response to THZ531, which covalently links to Cys1039 in CDK12 (Zhang et al, 2016). Nor was inhibition of PolII Ser2 phosphorylation by THZ531 prevented in the mutant cells (Fig. 5E). Note that THZ531 is not expected to promote degradation of CCNK or CDK12 proteins.

## Replacing Asp99 in CDK4 by Asn promotes resistance to CDK4/6i in MCF7 cells

Because of the clinical importance of CDK4/6i, we determined if substitution of this residue in CDK4 (D99) or CDK6 (D104) would affect response to ATP-competitive CDK4/6i. Indeed, co-crystal structures of CDK6 have revealed the importance of D104 for stabilisation of the piperazine ring in Palbociclib, Abemaciclib and Ribociclib (all ATP-competitive CDK4/6i) (Chen et al, 2016), while molecular docking studies imply a similar importance of the paralogous D99 for the inhibition of CDK4 by these drugs (Tadesse et al, 2017). The Aspartate to asparagine substitution impaired the binding of Palbociclib and Ribociclib to CDK4 and CDK6 in the NanoBRET in-cell assay (Fig. 6A,B). The mutation reduced apparent binding of Palbociclib to CDK4 and CDK6 by 20-fold. Binding to Ribociclib was reduced 8.7- and 14-fold for CDK4 and CDK6, respectively.

CDK6 is poorly expressed in MCF7 cells (Fig. EV5A), so is unlikely to be important for response to CDK4/6i in these cells. Therefore, we generated the CDK4 D99N mutation. Successful targeting of the CDK4 gene was confirmed by mutant-specific PCR (Fig. EV5B,C). Culturing with Palbociclib for 12 weeks enriched for cells with the D99N mutation. Palbociclib-resistant CDK4-D99N cells, plated at clonal densities, were used to obtain heterozygous (WT/D99N) and homozygous (D99N) mutant clones (Fig. EV5D). Immunoblotting shows that palbociclib-directed reduction in RB1

**A**

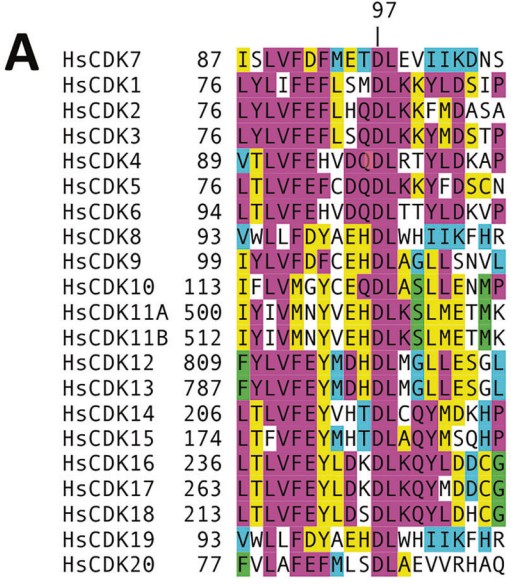

**B**

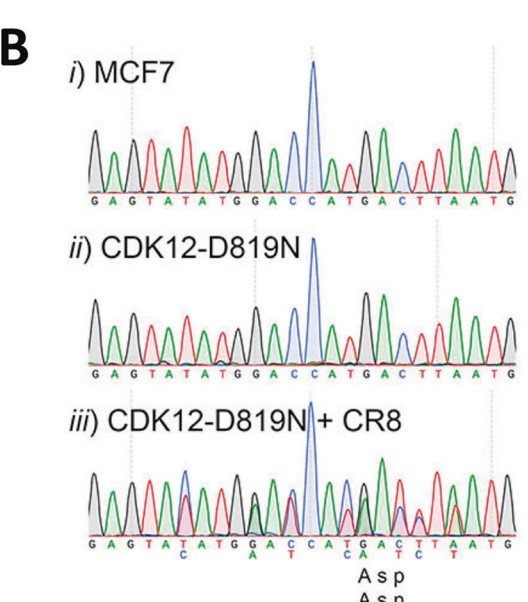

**C**

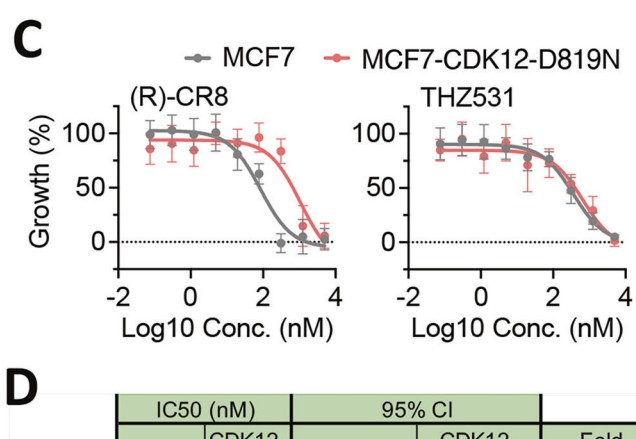

**D**

| | IC50 (nM) | | 95% CI | | |
|---|---|---|---|---|---|
| | MCF7 | CDK12-D819N | MCF7 | CDK12-D819N | Fold Resistance |
| (R)-CR8 | 84.5 | 1064 | 62.5-113 | 712.3-1629 | 12.6 |
| THZ531 | 356 | 656 | 251-506 | 323-1356 | 1.8 |

**E**

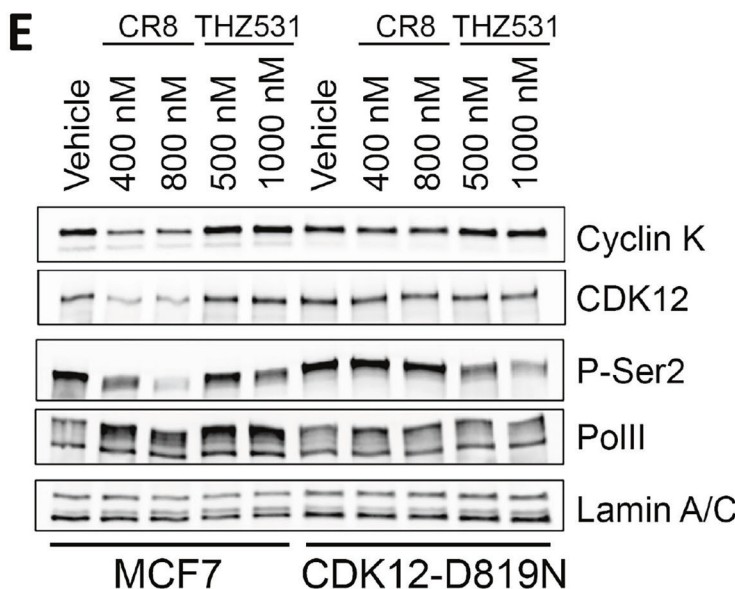

**Figure 5. The CDK7 Asp97 is conserved across all human CDKs.**

(A) Alignment of human CDKs demonstrates absolute conservation of Asp97 in human CDKs. (B) Sequencing chromatograms of gDNA prepared from MCF7-WT. Also shown are the gDNA sequences for (i) MCF7 cells, (ii) MCF7 following CRISPR-Cas9 mutagenesis and (iii) for gDNA prepared from cells after culturing for 4 weeks in the presence of 400 nM (R)-CR8. The positions of the silent changes, as well as the mutations in the codon encoding CDK12-D819, are shown, the mixed sequences at these positions being indicative of heterozygosity. (C) Growth assays were performed using 2× serial dilutions of the drugs. Error bars represent the standard errors of the mean (SEM) for $n = 2$ (THZ531) and $n = 3$ ((R)-CR8) independent experiments for MCF7 CDK12-WT and CDK12-D819N mutant cells. Each experiment contains measurements for six internal replicates. (D) Mean $IC_{50}$ values determined from the results in (C). (E) Immunoblotting of protein lysates from cells treated with (R)-CR8 or THZ531 for 6 h. An equal volume of DMSO was added to the vehicle controls. Source data are available online for this figure.

phosphorylation Ser807/811 was less pronounced in the homozygous mutant clone, CL1, than in the parental MCF7 cells (Figs. 6C and EV5E), suggesting that the D99N mutation reduces sensitivity to palbociclib. The mutation did not affect growth, with similar doubling times of 28.4 and 28.5 h for the homozygous mutant clone CL1 and heterozygous clone (CL4; MCF7-CDK4^WT/D99N), compared with MCF7 cells (28.2 h) in vehicle conditions (Figs. 6D and EV5F). MCF7 cells were growth inhibited by Palbociclib in a dose-dependent manner. In the presence of 1 μM Palbociclib, cells reached just 50% confluency at 250 h, doubling time increasing fivefold, to 138.9 (95% CI = 96.8–238.0) h. By contrast, mutant clones were substantially less sensitive to Palbociclib, with doubling times for 1 μM Palbociclib increasing just twofold to 55.3 (95% CI = 43.9–71.7) and 54.6 (95% CI = 48.4–62.1) h in heterozygous and homozygous mutant clones, respectively.

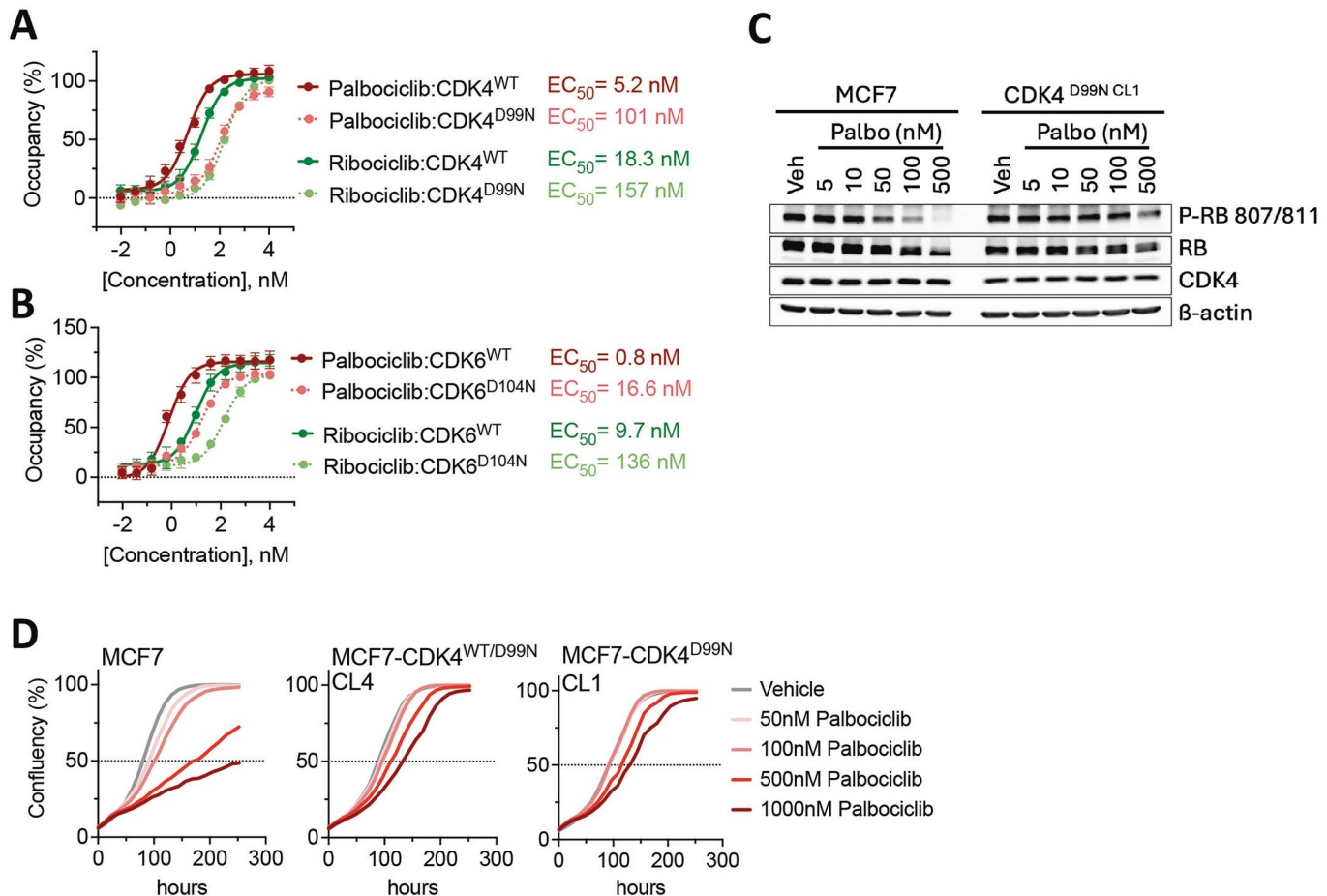

**Figure 6. Substitution of conserved aspartate residue in CDK4 and CDK6 reduces sensitivity to CDK4/6 inhibitors.**

(A, B) HEK293 cells transfected with Nano-luc CDK4 or CDK6 fusions, together with CCND1, were used in the NanoBRET assay for the displacement of the fluorescent tracer (K-10). Line graphs show the results of $n = 3$ independent experiments. Mean $EC_{50}$ values from the three experiments are listed. Error bars show SEM. (C, D) CRISPR-Cas9 gene editing was used to generate the D99N mutation in MCF7 cells. One heterozygous clone (CL4; MCF7-CDK4$^{WT/D99N}$) and a homozygous mutant clone (CL1; MCF7-CDK4$^{D99N}$) were cultured for 3 passages, over a period of 3–4 weeks in medium lacking Palbociclib for immunoblotting and growth studies. Error bars = SEM. (C) Cells were treated with Palbociclib for 48 h. Immunoblotting using antibodies for the indicated proteins/phosphorylation marks is shown for the homozygous CDK4-D99N mutant clone (CL1). (D) Growth was assessed in the presence of the indicated Palbociclib concentrations. The curves show mean growth over 250 h for $n = 3$ independent experiments. Source data are available online for this figure.

## Discussion

The critical roles of CDKs in controlling cell cycle progression and transcriptional processes, and the almost universal dysregulation of these processes in cancer, have inspired CDK inhibitor development. Most momentous has been the clinical success of CDK4/6i for ER+ breast cancer. CDK2 inhibitors INX-315 and PF-07104091 are being trialled in ER+ breast cancer patients following progression on CDK4/6i (Arora et al, 2023; Dietrich et al, 2024; Yap et al, 2023), while early-stage clinical trials for CDK7i (Coombes et al, 2023; Patel et al, 2018) and CDK9 (Xiao et al, 2023) reveal the potential of targeting transcriptional CDKs. Understanding mechanisms by which cancers can become resistant to this class of kinases is needed for the most suitable clinical deployment of CDK inhibitors. Our results show that acquired resistance to CDK7i can result from a mutation in the endogenous CDK7 gene that alters an aspartate that is absolutely conserved in all human CDKs. Cryo-EM structure determination revealed that

this mutation does not alter the CDK7 active site architecture and maintains binding to adenine-nucleotides but is indicative of greater heterogeneity of binding of CDK7i in the active site of CDK7i. The covalent linkage of THZ1 at cysteine 312 appears to allow maintenance of its position in the CDK7 active site, facilitating the continued observed inhibition of CDK7 activity. In-cell kinase binding assays demonstrate reduced binding of this CDK7-D97N mutant to ATP-competitive CDK7i but not to covalent CDK7i, consistent with continued sensitivity of mutant-expressing cells to covalent CDK7i. The CDK7-D97N does reduce kinase activity, but the lowering of activity is insufficient to impact normal cell growth, or phosphorylation of CDK7 targets PolII and CDK2 in mutant cells; this implies that expression levels of CDK7 are sufficiently high to mitigate for the reduction in activity.

CDK7-D97 is present in the hinge region connecting the N- and C-terminal kinase lobes, is absolutely conserved in all CDKs and structural studies reveal the involvement of this aspartate in binding of ATP-competitive CDK4, CDK6, CDK2 and CDK12

inhibitors (Chen et al, 2016; Dietrich et al, 2024; Tadesse et al, 2017; Yan et al, 2023). Targeted mutational substitution by asparagine in the endogenous CDK4 and CDK12 genes is also sufficient for resistance to inhibitors of these CDKs. Our results further indicate that the D>N mutation maintains sufficient kinase activity for cell growth, while reducing inhibitor binding. In the case of CDK4, another explanation for the lack of a proliferation block in mutant cells could be CDK6, since CDK4 and CDK6 have apparent redundant roles in G1 progression, at least in breast cancer cells. However, our analysis showed that CDK6 expression is low/absent in MCF7 cells, which were used for generating the CDK4 mutation.

A previous mutational scanning approach used ectopic expression of cDNA expression libraries containing almost all possible single amino acid substitutions for CDK4 and CDK6 in a melanoma cell line (Persky et al, 2020). Enrichment for mutations that reduce sensitivity to Palbociclib identified 56 and 66 amino acids in CDK4 and CDK6, respectively, suggesting that diverse mutations in these genes can drive resistance to CDK4/6i. Among the mutations that reduced sensitivity to Palbociclib was the aspartate studied here. Interestingly, this residue is conserved in ERK2, yet its mutation did not cause resistance to ATP-competitive ERK2 inhibitors (Brenan et al, 2016; Persky et al, 2020). Nor did mutation of the equivalent amino acid in BRAF, HER2 or ABL (in BCR-ABL) promote resistance to clinical inhibitors of those kinases. Given our demonstration here that mutation of this aspartate can cause resistance in CDK7, CDK4, CDK6 and CDK12, our results are indicative of the selectivity of this residue for the CDK class of kinases to ATP-competitive inhibitors.

Our results show that resistance due to mutations which reduce the binding of ATP-competitive CDK inhibitors may be avoided using irreversible covalent inhibitors. However, there are historical concerns with covalent drugs, due to the fact that the reactive groups required for covalent linking may cause off-target reactivity leading to hepatotoxicity and/or adverse immune responses to reactive intermediate formation (Erve, 2006); although limiting the use of problematic reactive groups in covalent drug design is now commonplace (Singh et al, 2011). Moreover, the residue bound by covalent kinase inhibitors can itself be subject to mutation that can prevent inhibitor binding but maintain kinase activity. In vitro, mutation of C312 that is bound by covalent CDK7i, to serine, prevented the binding of THZ1, while maintaining PolII phosphorylation activity (Kwiatkowski et al, 2014). In the clinic, the acquired BTK$^{C481S}$ mutation is a frequent cause of resistance to the covalent BTK inhibitor Ibrutinib in chronic lymphocytic leukaemia (CLL) patients (Woyach et al, 2017). Non-covalent BTK inhibitors

have consequently been developed and are effective for C481-mutant CLL patients (Mato et al, 2021).

Our study reveals a possible route to resistance involves acquired mutations that maintain kinase activity, albeit perhaps at lower levels, but which show greatly reduced inhibition by CDKi. Very few patients have so far been treated with CDK7i. CDK12i and CDK2i have only recently started in clinical trials, so it is too early for resistance mechanisms in patients to be clearly identified. We would also emphasise the point that biochemical studies often presage clinical discoveries. Another area we are working on is ER, which exemplifies this point. Biochemical studies identified amino acid substitutions that would be capable of causing resistance to hormone therapies (Weis et al, 1996; White et al, 1997) many years prior to the discovery of the very same mutations in ER+ metastatic breast cancer patients (Li et al, 2013; Robinson et al, 2013; Toy et al, 2013). However, in the case of CDK4/6i, mutational analysis of patient samples has been undertaken to elucidate resistance mechanisms. These studies have not reported mutations in CDK4 or CDK6, although may reflect the ready emergence of resistance through other mechanisms, including CDK6 amplification/overexpression, RB1 loss/mutational inactivation, cyclin E1/E2 amplification leading to elevated CDK2 activity and elevated growth factor receptor signalling (Morrison et al, 2024).

In conclusion, our findings reveal the intriguing possibility that mutations in the CDK7 gene provide a route to acquired resistance to CDK7i in cancer. Even though CDK7i are moving to advanced clinical trials, biomarkers of patient response and mechanisms of acquired resistance remain unclear. By discovering one route to CDK7i resistance, we identify the importance of patient monitoring for the emergence of CDK7 gene mutations, for example, by assessment of circulating cell-free DNA. Although we have focussed here on the one amino acid residue, our data confirm the possibility of resistance to other CDKi through mutation of the targeted CDK and so support monitoring for acquired mutations in the relevant CDKs in patients being treated with other CDK inhibitors.

# Methods

## Experimental models

Human LNCaP cells were purchased from ATCC, human 22Rv1 cells were provided by Dr. S. Omarjee and Dr. J. Carroll (CRUK Cambridge). Human MCF7-luc cells were purchased from Cell

**Reagents and tools table**

| Reagent/resource | Reference or source | Identifier or catalogue number |
|---|---|---|
| **Experimental models** | | |
| LNCaP | ATCC | Cat#CRL-1740 |
| 22Rv1 | DRs S. Omarjee and J. Carroll (CRUK Cambridge) | |
| MCF7 Luc | Cell Biolabs | Cat#AKR-234 |
| High Five | Thermo Fisher Scientific | Cat#B85502 |
| Sf9 | Thermo Fisher Scientific | Cat#11496015 |

| Reagent/resource | Reference or source | Identifier or catalogue number |
|---|---|---|
| **Recombinant DNA** | | |
| pNLF1-N NanoLuc Protein Fusion Vector | Promega | Cat#N1351 |
| Rc/CMV cyclin D1 HA | Addgene | Cat#8948 |
| 438Sn-CDK7 D97N | This study | |
| 438C-MAT1Δ219-cyclin H | Greber et al (2020) | |
| **Antibodies** | | |
| P-CDK2 (Thr160) | CST | Cat#2651S |
| CDK2 | CST | Cat#2546S |
| P-Rb (Ser807/811) | CST | Cat#9308S |
| Rb | CST | Cat#9309S |
| CDK7 | CST | Cat#2916S |
| CDK12 | CST | Cat#11973S |
| P-Ser2 PolII | Abcam | Cat#ab5095 |
| P-Ser5 PolII | Abcam | Cat#ab5131 |
| P-Ser7 PolII | Abcam | Cat#ab252853 |
| PolII | Abcam | Cat#ab817 |
| β-actin | Abcam | Cat#ab6276 |
| Goat Anti-mouse Immunoglobulins/HRP | Dako Agilent | Cat#P0448 |
| Goat Anti-Rabbit Immunoglobulins/HRP | Dako Agilent | Cat#P0447 |
| IRDye 800CW Goat anti-Rabbit IgG Secondary Antibody | LICORbio | Cat#926-32211 |
| **Oligonucleotides and other sequence-based reagents** | | |
| Alt-R CRISPR-Cas9 tracrRNA | IDT | Cat#1072533 |
| IDT Alt-R CRISPR-Cas9 crRNA | IDT (in this study) | **CDK7:** TTTCCATAAAATCAAAGACA<br>**CDK4:** TGGAACTTTATCCAAGTAAG<br>**CDK12:** CTAGCAGTCCCATTAAGTCA |
| HDR donor template | IDT (in this study) | **CDK7 D97N HDR donor:**<br>ATTTTTTAACTTTGCAG<br>CTCCTTGATGCTTTT<br>GGACATAA<br>GAGTAACATATCT<br>TTGGTATTCG<br>ACTTTATGGAAACT<br>AATCTAGAGGTA<br>AGATTAAGATTC<br>CTGGAGAAATTCCTTTG<br>**CDK4 D99N HDR donor:**<br>CCCGAACTGACCGGGA<br>GATCAAGGTAACCCTGGTGTTTGA<br>ACACGTCGATCAGAATCTCCGCACCTAT<br>CTGGACAAGGCAC<br>CCCCACCAGGCTTGCCAGCCGAAA<br>**CDK12 D819N HDR donor:**<br>TTCGTCTTTATGTAGGTGCCTTTTAC<br>CTTGTATTTGAGTAC<br>ATGGATCACAATCTTATGGGACTGCT<br>AGAATCTGGTTTGGT<br>GCACTTTTCTGAGGACCA |

| Reagent/resource | Reference or source | Identifier or catalogue number |
|---|---|---|
| PCR primers | IDT (in this study) | **CDK7 D97N knock-in specific:**<br>**Forward:** GAGTAACATATCTTTGGTATTCGACTT<br>**Reverse:** TACTACCAGTATCTTCTTGGGTAGA<br>**CDK7 locus:**<br>**Forward:** TTGCCTCAGTTGCTATGATACT<br>**Reverse:** ACGTCAACTAGCTATGGGAACT<br>**CDK7 ORF (from cDNA):**<br>**Forward:** GGGGATACTAAAGCGACGGA<br>**Reverse:** TTTTGGCTATTTCCCTCAGTAGT<br>**CDK4 D99N knock-in specific:**<br>**Forward:** ACACGTCGATCAGAATCTCC<br>**Reverse:** ATCCACCTCTCAATGCCTAC<br>**CDK4 locus:**<br>**Forward:** AGGTGGGGTGTGATGATCTG<br>**Reverse:** AAGGGGAGGTACAGATGCAC<br>**CDK12 D819N knock-in specific:**<br>**Forward:** GGACTTGAGGCATT<br>GTTATTT<br>**Reverse:** CCATAAGATTGTGATCCATG<br>**CDK12 locus:**<br>**Forward:** CGCCCAGCCACAGAAGATTA<br>**Reverse:** GAGGAGAAGAGGAAAGTGCTTAA<br>**CDK7 D97N mutagenesis:**<br>**Forward:** GGAGACTAACCTGGAAGTGATCATC<br>**Reverse:** TCCAGGTTAGTCTCCATGAAGTCG |
| gBlocks for CDK HiFi DNA assembly into XhoI digested pNLF1-N | IDT (in this study) | **CDK7 WT gBlock:**<br>ACCGGCTGGCGGCTGTGCGAACGCATTCTGGCGGGCAGCA<br>GTGGAGCTATTGCAGCTCTGGACGTGAAGTCTCGGGCAA<br>AGCGTTATGAGAAGCTGGACTTCCTTGGGGAGGGACAGT<br>TTGCCACCGTTTACAAGGCCAGAGATAAGAACACCAACC<br>AAATTGTCGCCATTAAGAAAATCAAACTTGGACATAGAT<br>CAGAAGCTAAAGATGGTATAAATAGAACCGCCTTAAGA<br>GAGATAAAATTATTACAGGAGCTAAGTCATCCAAATATA<br>ATTGGTCTCCTTGATGCTTTTGGACATAAATCTAATATTA<br>GCCTTGTCTTTGATTTTATGGAAACTGATCTAGAGGTTAT<br>AATAAAGGATAATAGTCTTGTGCTGACACCATCACACAT<br>CAAAGCCTACATGTTGATGACTCTTCAAGGATTAGAATA<br>TTTACATCAACATTGGATCCTACATAGGGATCTGAAACC<br>AAACAACTTGTTGCTAGATGAAAATGGAGTTCTAAAACT<br>GGCAGATTTTGGCCTGGCCAAATCTTTTGGGAGCCCCAA<br>TAGAGCTTATACACATCAGGTTGTAACCAGGTGGTATCG<br>GGCCCCCGAGTTACTATTTGGAGCTAGGATGTATGGTGT<br>AGGTGTGGACATGTGGGCTGTTGGCTGTATATTAGCAGA<br>GTTACTTCTAAGGGTTCCTTTTTTGCCAGGAGATTCAGA<br>CCTTGATCAGCTAACAAGAATATTTGAAACTTTGGGCAC<br>ACCAACTGAGGAACAGTGGCCGGACATGTGTAGTCTTCC<br>AGATTATGTGACATTTAAGAGTTTCCCTGGAATACCTTTG<br>CATCACATCTTCAGTGCAGCAGGAGACGACTTACTAGAT<br>CTCATACAAGGCTTATTCTTATTTAATCCATGTGCTCGAA<br>TTACGGCCACACAGGCACTGAAAATGAAGTATTTCAGTA<br>ATCGGCCAGGGCCAACACCTGGATGTCAGCTGCCAAGA<br>CCAAACTGTCCAGTGGAAACCTTAAAGGAGCAATCAAA<br>TCCAGCTTTGGCAATAAAAAGGAAAAGAACAGAGGCCT<br>TAGAACAAGGAGGATTGCCCAAGAAACTAATTTTTTAAC<br>TCGAGCGGCGCGATCGCTTCCGAATTCAGAGCTCAACC<br>GCG<br>**CDK7 D97N gBlock:**<br>ACCGGCTGGCGGCTGTGCGAACGCATTCTGGCGGGCAGC<br>AGTGGAGCTATTGCAGCTCTGGACGTGAAGTCTCGGGCAA<br>AGCGTTATGAGAAGCTGGACTTCCTTGGGGAGGGACAGTT<br>TGCCACCGTTTACAAGGCCAGAGATAAGAACACCAACCA<br>AATTGTCGCCATTAAGAAAATCAAACTTGGACATAGATCA<br>GAAGCTAAAGATGGTATAAATAGAACCGCCTTAAGAGAG<br>ATAAAATTATTACAGGAGCTAAGTCATCCAAATATAATTG<br>GTCTCCTTGATGCTTTTGGACATAAATCTAATATTAGCCTT<br>GTCTTTGATTTTATGGAAACTAATCTAGAGGTTATAATAA<br>AGGATAATAGTCTTGTGCTGACACCATCACACATCAAAGC<br>CTACATGTTGATGACTCTTCAAGGATTAGAATATTTACATC<br>AACATTGGATCCTACATAGGGATCTGAAACCAAACAACTTG |

| Reagent/resource | Reference or source | Identifier or catalogue number |
| --- | --- | --- |
| | | TTGCTAGATGAAAATGGAGTTCTAAAACTGGCAGATTTTGG<br>CCTGGCCAAATCTTTTGGGAGCCCCAATAGAGCTTATACAC<br>ATCAGGTTGTAACCAGGTGGTATCGGGCCCCCGAGTTACTA<br>TTTGGAGCTAGGATGTATGGTGTAGGTGTGGACATGTGGGC<br>TGTTGGCTGTATATTAGCAGAGTTACTTCTAAGGGTTCCTTT<br>TTTGCCAGGAGATTCAGACCTTGATCAGCTAACAAGAATAT<br>TTGAAACTTTGGGCACACCAACTGAGGAACAGTGGCCGGAC<br>ATGTGTAGTCTTCCAGATTATGTGACATTTAAGAGTTTCCCT<br>GGAATACCTTTGCATCACATCTTCAGTGCAGCAGGAGACGA<br>CTTACTAGATCTCATACAAGGCTTATTCTTATTTAATCCATG<br>TGCTCGAATTACGGCCACACAGGCACTGAAAATGAAGTATT<br>TCAGTAATCGGCCAGGGCCAACACCTGGATGTCAGCTGCCA<br>AGACCAAACTGTCCAGTGGAAACCTTAAAGGAGCAATCAA<br>ATCCAGCTTTGGCAATAAAAAGGAAAAGAACAGAGGCCTT<br>AGAACAAGGAGGATTGCCCAAGAAACTAATTTTTTAACTC<br>GAGCGGCGCGATCGCTTCCGAATTCAGAGCTCAACCGCG<br>**CDK4 WT gBlock:**<br>ACCGGCTGGCGGCTGTGCGAACGCATTCTGGCGGGCAGCA<br>GTGGAGCTATTGCAGCTACCTCTCGATATGAGCCAGTGGCT<br>GAAATTGGTGTCGGTGCCTATGGGACAGTGTACAAGGCCC<br>GTGATCCCCACAGTGGCCACTTTGTGGCCCTCAAGAGTGTG<br>AGAGTCCCCAATGGAGGAGGAGGTGGAGGAGGCCTTCCCA<br>TCAGCACAGTTCGTGAGGTGGCTTTACTGAGGCGACTGGA<br>GGCTTTTGAGCATCCCAATGTTGTCCGGCTGATGGACGTCT<br>GTGCCACATCCCGAACTGACCGGGAGATCAAGGTAACCCT<br>GGTGTTTGAGCATGTAGACCAGGACCTAAGGACATATCTG<br>GACAAGGCACCCCCACCAGGCTTGCCAGCCGAAACGATCA<br>AGGATCTGATGCGCCAGTTTCTAAGAGGCCTAGATTTCCTT<br>CATGCCAATTGCATCGTTCACCGAGATCTGAAGCCAGAGAA<br>CATTCTGGTGACAAGTGGTGGAACAGTCAAGCTGGCTGACT<br>TTGGCCTGGCCAGAATCTACAGCTACCAGATGGCACTTACA<br>CCCGTGGTTGTTACACTCTGGTACCGAGCTCCCGAAGTTCTT<br>CTGCAGTCCACATATGCAACACCTGTGGACATGTGGAGTG<br>TTGGCTGTATCTTTGCAGAGATGTTTCGTCGAAAGCCTCTC<br>TTCTGTGGAAACTCTGAAGCCGACCAGTTGGGCAAAATCTT<br>TGACCTGATTGGGCTGCCTCCAGAGGATGACTGGCCTCGAG<br>ATGTATCCCTGCCCCGTGGAGCCTTTCCCCCCAGAGGGCCC<br>CGCCCAGTGCAGTCGGTGGTACCTGAGATGGAGGAGTCGGG<br>AGCACAGCTGCTGCTGGAAATGCTGACTTTTAACCCACACA<br>AGCGAATCTCTGCCTTTCGAGCTCTGCAGCACTCTTATCTAC<br>ATAAGGATGAAGGTAATCCGGAGTGACTCGAGCGGCGCGA<br>TCGCTTCCGAATTCAGAGCTCAACCGCG<br>**CDK4 D99N gBlock:**<br>ACCGGCTGGCGGCTGTGCGAACGCATTCTGGCGGGCAGCAG<br>TGGAGCTATTGCAGCTACCTCTCGATATGAGCCAGTGGCTG<br>AAATTGGTGTCGGTGCCTATGGGACAGTGTACAAGGCCCGT<br>GATCCCCACAGTGGCCACTTTGTGGCCCTCAAGAGTGTGAG<br>AGTCCCCAATGGAGGAGGAGGTGGAGGAGGCCTTCCCATC<br>AGCACAGTTCGTGAGGTGGCTTTACTGAGGCGACTGGAGG<br>CTTTTGAGCATCCCAATGTTGTCCGGCTGATGGACGTCTGT<br>GCCACATCCCGAACTGACCGGGAGATCAAGGTAACCCTGG<br>TGTTTGAGCATGTAGACCAGAACCTAAGGACATATCTGGA<br>CAAGGCACCCCCACCAGGCTTGCCAGCCGAAACGATCAAG<br>GATCTGATGCGCCAGTTTCTAAGAGGCCTAGATTTCCTTCA<br>TGCCAATTGCATCGTTCACCGAGATCTGAAGCCAGAGAAC<br>ATTCTGGTGACAAGTGGTGGAACAGTCAAGCTGGCTGACT<br>TTGGCCTGGCCAGAATCTACAGCTACCAGATGGCACTTACA<br>CCCGTGGTTGTTACACTCTGGTACCGAGCTCCCGAAGTTCTT<br>CTGCAGTCCACATATGCAACACCTGTGGACATGTGGAGTGT<br>TGGCTGTATCTTTGCAGAGATGTTTCGTCGAAAGCCTCTCTT<br>CTGTGGAAACTCTGAAGCCGACCAGTTGGGCAAAATCTTTG<br>ACCTGATTGGGCTGCCTCCAGAGGATGACTGGCCTCGAGAT<br>GTATCCCTGCCCCGTGGAGCCTTTCCCCCCAGAGGGCCCCG<br>CCCAGTGCAGTCGGTGGTACCTGAGATGGAGGAGTCGGGA<br>GCACAGCTGCTGCTGGAAATGCTGACTTTTAACCCACACAA<br>GCGAATCTCTGCCTTTCGAGCTCTGCAGCACTCTTATCTACA<br>TAAGGATGAAGGTAATCCGGAGTGACTCGAGCGGCGCGA<br>TCGCTTCCGAATTCAGAGCTCAACCGCG<br>**CDK6 WT gBlock:**<br>ACCGGCTGGCGGCTGTGCGAACGCATTCTGGCGGGCAGCA<br>GTGGAGCTATTGCAGAGAAGGACGGCCTGTGCCGCGCTGAC |

| Reagent/resource | Reference or source | Identifier or catalogue number |
|---|---|---|
| | | CAGCAGTACGAATGCGTGGCGGAGATCGGGGAGGGCGCCT<br>ATGGGAAGGTGTTCAAGGCCCGCGACTTGAAGAACGGAGG<br>CCGTTTCGTGGCGTTGAAGCGCGTGCGGGTGCAGACCGGC<br>GAGGAGGGCATGCCGCTCTCCACCATCCGCGAGGTGGCGGT<br>GCTGAGGCACCTGGAGACCTTCGAGCACCCCAACGTGGTCA<br>GGTTGTTTGATGTGTGCACAGTGTCACGAACAGACAGAGAA<br>ACCAAACTAACTTTAGTGTTTGAACATGTCGATCAAGACTT<br>GACCACTTACTTGGATAAAGTTCCAGAGCCTGGAGTGCCCA<br>CTGAAACCATAAAGGATATGATGTTTCAGCTTCTCCGAGGT<br>CTGGACTTTCTTCATTCACACCGAGTAGTGCATCGCGATCT<br>AAAACCACAGAACATTCTGGTGACCAGCAGCGGACAAATA<br>AAACTCGCTGACTTCGGCCTTGCCCGCATCTATAGTTTCCA<br>GATGGCTCTAACCTCAGTGGTCGTCACGCTGTGGTACAGA<br>GCACCCGAAGTCTTGCTCCAGTCCAGCTACGCCACCCCCGT<br>GGATCTCTGGAGTGTTGGCTGCATATTTGCAGAAATGTTT<br>CGTAGAAAGCCTCTTTTTCGTGGAAGTTCAGATGTTGATCA<br>ACTAGGAAAAATCTTGGACGTGATTGGACTCCCAGGAGAA<br>GAAGACTGGCCTAGAGATGTTGCCCTTCCCAGGCAGGCTTT<br>TCATTCAAAATCTGCCCAACCAATTGAGAAGTTTGTAACAG<br>ATATCGATGAACTAGGCAAAGACCTACTTCTGAAGTGTTTG<br>ACATTTAACCCAGCCAAAAGAATATCTGCCTACAGTGCCCT<br>GTCTCACCCATACTTCCAGGACCTGGAAAGGTGCAAAGAA<br>AACCTGGATTCCCACCTGCCGCCCAGCCAGAACACCTCGGA<br>GCTGAATACAGCCTGACTCGAGCGGCGCGATCGCTTCCGA<br>ATTCAGAGCTCAACCGCG<br>**CDK6 D104N gBlock:**<br>ACCGGCTGGCGGCTGTGCGAACGCATTCTGGCGGGCAGCAG<br>TGGAGCTATTGCAGAGAAGGACGGCCTGTGCCGCGCTGACC<br>AGCAGTACGAATGCGTGGCGGAGATCGGGGAGGGCGCCTA<br>TGGGAAGGTGTTCAAGGCCCGCGACTTGAAGAACGGAGGC<br>CGTTTCGTGGCGTTGAAGCGCGTGCGGGTGCAGACCGGCGA<br>GGAGGGCATGCCGCTCTCCACCATCCGCGAGGTGGCGGTGC<br>TGAGGCACCTGGAGACCTTCGAGCACCCCAACGTGGTCAGG<br>TTGTTTGATGTGTGCACAGTGTCACGAACAGACAGAGAAAC<br>CAAACTAACTTTAGTGTTTGAACATGTCGATCAA<br>AACTTGACCACTTACT<br>TGGATAAAGTTCCAGAGCCTGGAGTGCCCACTGAAACC<br>ATAAAGGATATGATGTTTCAGCTTCTCCGAGGTCTGGACTTT<br>CTTCATTCACACCGAGTAGTGCATCGCGATCTAAAACCACA<br>GAACATTCTGGTGACCAGCAGCGGACAAATAAAACTCGCTG<br>ACTTCGGCCTTGCCCGCATCTATAGTTTCCAGATGGCTCTA<br>ACCTCAGTGGTCGTCACGCTGTGGTACAGAGCACCCGAAG<br>TCTTGCTCCAGTCCAGCTACGCCACCCCCGTGGATCTCTGG<br>AGTGTTGGCTGCATATTTGCAGAAATGTTTCGTAGAAAGC<br>CTCTTTTTCGTGGAAGTTCAGATGTTGATCAACTAGGAAA<br>AATCTTGGACGTGATTGGACTCCCAGGAGAAGAAGACTG<br>GCCTAGAGATGTTGCCCTTCCCAGGCAGGCTTTTCATTCA<br>AAATCTGCCCAACCAATTGAGAAGTTTGTAACAGATATCGA<br>TGAACTAGGCAAAGACCTACTTCTGAAGTGTTTGACATTT<br>AACCCAGCCAAAAGAATATCTGCCTACAGTGCCCTGTCTC<br>ACCCATACTTCCAGGACCTGGAAAGGTGCAAAGAAAACCT<br>GGATTCCCACCTGCCGCCCAGCCAGAACACCTCGGAGCTG<br>AATACAGCCTGACTCGAGCGGCGCGATCGCTTCCGAATTC<br>AGAGCTCAACCGCG |

**Chemicals, enzymes and other reagents**

| Reagent/resource | Reference or source | Identifier or catalogue number |
|---|---|---|
| Samuraciclib | Carrick Therapeutics/<br>MCE | Gifted<br>Cat#S8722 |
| SY-5609 | MCE | Cat# HY-138293 |
| LDC4297 | Selleckchem | Cat#S7992 |
| THZ1 | ApexBio/<br>Merck Millipore | Cat#A8882<br>Cat#532372 |
| SY-1365 | MCE | Cat#HY-128587 |
| YKL-5-124 | MCE | Cat#HY-101257 |
| (R)-CR8 | Tocris | Cat#3605 |
| THZ531 | MCE | Cat# HY-103618 |
| Palbociclib | Pfizer | Gifted |

| Reagent/resource | Reference or source | Identifier or catalogue number |
|---|---|---|
| Ribociclib | Selleckchem | Cat#S7440 |
| Alt-R S.p. Cas9 Nuclease V3 | IDT | Cat#1081058 |
| Alt-R HDR Enhancer | IDT | Cat#10007910 |
| NanoBRET TE Intracellular Kinase Assay K-10 | Promega | Cat#N2642 |
| SE Cell Line 4D-Nucleofector X Kit | Lonza | Cat#V4XC-1012 |
| XhoI1 | NEB | Cat#R0146S |
| HiFi DNA Assembly Cloning Kit | NEB | Cat#E5520S |
| ExoSAP-IT | Thermo Fisher | Cat#78201.1 |
| PureLink Genomic DNA Mini Kit | Invitrogen | Cat#K182002 |
| RNeasy Mini Kit | Qiagen | Cat#74106 |
| REDTaq ReadyMix PCR Reaction Mix | Sigma-Aldrich | Cat#R2523 |
| Q5 High-Fidelity 2X Master Mix | NEB | Cat#M0492S |
| RevertAid Reverse Transcriptase | Thermo Scientific | Cat#EP0441 |
| cOmplete, Mini, EDTA-free Protease Inhibitor Cocktail | Roche | Cat#11836170001 |
| Phosphatase Inhibitor Mini Tablets | Thermo Scientific | Cat#A32957 |
| RIPA Buffer | Sigma-Aldrich | Cat#R0278 |
| BCA Protein Assay Kits | Thermo Scientific | Cat#23225 |
| Strep-Tactin Superflow Plus | Qiagen | Cat#30004 |
| Ni-NTA Superflow | Qiagen | Cat#30410 |
| CellFectin II Reagent | Thermo Scientific | Cat#10362100 |
| ATPγS | Sigma-Aldrich | Cat#A1388 |
| KOD One PCR Master Mix Blue | Sigma-Aldrich | Cat#KMM-210NV |
| In-Fusion Snap Assembly Master Mix | Takara Bio | Cat#638948 |
| **Software** | | |
| SnapGene | https://www.snapgene.com | |
| IncuCyte Zoom 2018A | https://downloads.essenbioscience.com/get/incucyte-zoom-2018a-controller-update | |
| Prism 10 | https://www.graphpad.com | |
| CryoSPARC Live | https://cryosparc.com/live | |
| RELION 5.0 | https://relion.readthedocs.io/en/release-5.0/index.html | |
| Phenix-1.21-5207 | https://phenix-online.org/ | |
| Coot 0.9.6 | https://www2.mrc-lmb.cam.ac.uk/personal/pemsley/coot/ | |
| UCSF ChimeraX-1.8 | https://www.cgl.ucsf.edu/chimerax/ | |

| Reagent/resource | Reference or source | Identifier or catalogue number |
|---|---|---|
| EPU | Thermo Fisher Scientific https://www.thermofisher.com/uk/en/home/electron-microscopy/products/software-em-3d-vis/epu-software.html | |
| ImageJ | https://imagej.net/ij/download.html | |
| Other | | |
| Krios G3i | Thermo Fisher Scientific | |
| Falcon 4i | Thermo Fisher Scientific | |
| Selectris X | Thermo Fisher Scientific | |
| K3-Bioquantum | Gatan | |
| Vitrobot Mk IV | Thermo Fisher Scientific | |
| Tergeo EM | PIE Scientific | |
| UltrAuFoil R1.2/1.3 | Quantifoil Microtools | Cat#N1-A14nAu30-01 |

Biolabs Inc., USA and were stably transduced with pTRIP-SFFV-EGFP-NLS (Addgene#86677). All cell lines were cultured according to supplier recommendations. Cell lines were regularly tested for mycoplasma. Samuraciclib (aka ICEC0942 (Patel et al, 2018)) was provided by Dr A Bahl (Carrick Therapeutics). THZ1 and (R)-CR8 from ApexBio and Tocris Bioscience, respectively. SY1365, SY5609, YKL-5-124 and THZ531 were obtained from MedChemExpress. Palbociclib was kindly provided by Pfizer Inc. LDC4297 and Ribociclib were purchased from Selleck Biochemicals. All drugs were solubilised in DMSO.

## Development of samuraciclib-resistant 22Rv1 cells

To generate samuraciclib-resistant 22Rv1 (SamR) cells, $6 \times 10^6$ 22Rv1 cells were seeded into a T75 flask. The following day, samuraciclib was added to the culture medium at a final concentration of 500 nM. Cells were incubated with samuraciclib for 1 month, with the medium replaced every 3 days. After 1 month, cells were passaged and maintained in medium containing 500 nM samuraciclib. Cells were routinely passaged upon reaching confluency and continuously cultured in the presence of samuraciclib for an additional 3 months. Cells were cultured in samuraciclib-free medium for at least 3 weeks prior to challenging with CDK7i.

## Growth assays

To determine the $IC_{50}$ of drug treatments on growth, cells were seeded in 96-well culture plates. Two days after seeding, the cells were treated with increasing concentrations of drugs and incubated for 4 days. Cell number was determined using the SRB assay (Skehan et al, 1990), and $IC_{50}$ values were calculated using non-linear regression curve fitting in GraphPad Prism v10. For all other growth experiments, cells were seeded into 96-well plates, and appropriate drugs were added 2 days after seeding. Cells were then incubated in an IncuCyte Zoom live-cell imager, with cell confluency measured every 6 h. Cell doubling times were determined by non-linear curve fitting (exponential growth equation) in GraphPad Prism v10.

## Generation of CRISPR-Cas9 knock-in cells

Alt-R$^{TM}$ S.p. Cas9 nuclease V3, Alt-R CRISPR-Cas9 crRNA, TracrRNA and the donor templates were obtained from Integrated DNA Technologies. In all, 1 µM of RNP complexes and 2.5 µM of HDR donor template were transfected into target cells by a Lonza 4d-Nucleofector using a Lonza SE cell line 4D-nucleofector kit with DS-109 pulse programme. Cells were treated with 0.7 µM of Alt-R HDR Enhancer V2 to improve HRD efficiency. Sequences of the CRISPRs in the crRNA and the donor templates are shown in the Reagents and tools table. Donor templates were designed to include non-coding substitutions to aid genome DNA PCR for identifying knock-in events and to facilitate cloning; these substitutions are noted in the Reagents and tools table.

## Genomic DNA and RNA preparation, RT-PCR, Sanger sequencing and RNA-sequencing

Genomic DNA was prepared by using an Invitrogen PureLink Genomic DNA Mini kit, according to the manufacturer's methods. Total RNA was prepared 72 h after seeding of 300,00 cells in individual wells of 6-well plates and culturing in RPMI supplemented with 10% FCS, using the RNeasy Mini Kit (Qiagen 74106). RNA (1–2 µg) was used for cDNA synthesis using a recombinant M-MuLV Reverse Transcriptase (ThermoFisher). RT-qPCR reactions were carried out using Fast SYBR Green Master Mix (ThermoFisher Scientific, 4385616). RNA-seq, read analysis, alignment and subsequent bioinformatics analysis were undertaken for six replicate RNA samples on the Illumina Novaseq 6000 platform by Novogene, Cambridge, UK.

## Immunoblotting

Whole-cell lysates were prepared in RIPA buffer (Sigma-Aldrich), supplemented with protease and phosphatase inhibitor cocktails (Roche). Protein sample concentrations were determined using the Pierce BCA Protein Assay kit (ThermoFisher Scientific). In all, 10–20 µg of total protein was loaded for each sample and resolved

on 4–15% Mini-PROTEAN TGX Precast Protein Gels (BIO-RAD). Immunoblotting was performed as described (Patel et al, 2018), using antibodies diluted 1:1000 for P-CDK2 (Thr160; 2561S), CDK2 (2546S), P-Rb (Ser807/811; 9308), Rb (9309), CDK7 (2916) and CDK12 (11973S), from Cell Signalling Technology. PolII CTD phosphorylation antibodies (P-Ser2 (ab5095) and P-Ser5 (ab5131), used at 1:10,000 dilution), as well as total PolII (ab817; 1:2000 dilution) and β-actin (ab6276; 1:100,000 dilution) were purchased from Abcam. CCNK antibody (Thermofisher A301-939A) was used at a 1:1000 dilution. Band intensities were quantified using ImageJ (Schneider et al, 2012).

## NanoBRET assay

CDK4, CDK6 and CDK7 and the respective mutant sequences, obtained as synthesised DNA fragments from IDT (Leuven), were cloned by HiFi DNA assembly (New England Biolabs) into the pNLF1-N NanoLuc protein fusion vector (Promega). HEK293 cells, transiently transfected with NanoLuc-CDK fusion constructs, were used for activity determination, utilising the NanoBRET Target Engagement Intracellular Kinase Assay. A CCND1 expression construct was co-transfected with the NanoLuc-CDK4 and NanoLuc-CDK6 fusion constructs. The recommended cell-permeable energy transfer probe, K-10, was used to assess target engagement, as detailed in the manufacturer's methods (Promega). Tracer affinity for CDKs was obtained using an 11-point, twofold dilution series starting from 1 µM K-10. The BRET ratio (mBU) was calculated as acceptor emission (590–610 nm)/donor emission (410–490 nm) × 1000. The mBU values were plotted against the energy probe concentrations to determine maximal binding ($B_{max}$) and $K_d$ values. BRET signal competition by CDK inhibitors was used to determine compound binding to WT and mutant CDKs. As maximal tracer binding ($B_{max}$) was reached with approximately 500 nM K-10 for CDK7 and 125 nM for CDK4/CDK6, the tracer was used at these concentrations for inhibitor assessment. Inhibitor binding was determined using a fourfold dilution series. mBU values were plotted against test compound concentrations in the presence of K10 tracer. EC$_{50}$ values were determined using nonlinear regression curve-fitting. Test compound occupancy (%) was calculated using the equation $[1 - (x - z)/(y - z)] \times 100$, where $x$ = mBU in the presence of the test compound and K-10 tracer, $y$ = mBU with K-10 tracer and vehicle control, and $z$ = mBU in the absence of the K-10 tracer or test compound (Wells et al, 2020).

## CAK expression and purification

CAK WT and CAK-D97N complexes were expressed in insect cells as the MAT1Δ219-cyclin H-CDK7 (CAK-MAT1Δ219) complex reported previously, which lacks the N-terminal 219 residues of MAT1 (Cushing et al, 2024; Greber et al, 2020). A construct of Strep-CDK7 (D97N) in a 438-series vector suitable for insect cell expression was generated by mutagenesis of the WT plasmid by PCR using mutagenic primers. Bacmid DNA generated from this construct and a 438-series construct encoding His$_6$-MBP-MAT1Δ219-cyclin H was used to infect Sf9 (*Spodoptera frugiperda*) cells to produce baculoviruses that could be used for co-expression in High Five (*Trichoplusia ni*) cells. Two 500 ml High Five cultures were each supplemented with 10 ml of each virus and incubated at

27 °C for 72 h. Cells were harvested by spinning at 1500 × *g* for 20 min at 4 °C, then pellets were snap frozen in liquid nitrogen for storage at −80 °C.

Pellets were resuspended in purification buffer (40 mM HEPES pH 7.9, 250 mM KCl, 10% (v/v) glycerol, 2 mM MgCl$_2$, 5 mM β-mercaptoethanol) supplemented with 10 mM imidazole, protease inhibitors and DNaseI. Resuspended pellets were combined, lysed by sonication and the lysate clarified by spinning at 18,000 rpm for 30 min at 4 °C. The clarified lysate was incubated with equilibrated Ni-NTA Superflow beads for 45 min at 4 °C. The beads were washed with purification buffer supplemented with 25 mM imidazole before elution with purification buffer supplemented with 300 mM imidazole. The eluted fractions were incubated with equilibrated StrepTactin Superflow Plus beads (Qiagen) for 45 min at 4 °C. The beads were washed with purification buffer before elution with purification buffer supplemented with 10 mM desthiobiotin. The eluted fractions were combined and incubated with His$_6$-TEV protease at a TEV:protein ratio of 1:10 by mass for 2 h at room temperature (RT) to cleave the tags. The TEV was then removed by incubating the sample with Ni-NTA beads at RT for 30 min. The flow-through was concentrated to approx. 800 µl and snap frozen in two aliquots for storage at −80 °C. Gel filtration was performed on one thawed aliquot using a Superdex 200 Increase 10/300 column with gel filtration buffer (20 mM HEPES pH 7.9, 200 mM KCl, 5% (v/v) glycerol, 2 mM MgCl$_2$, 5 mM β-mercaptoethanol). Selected fractions were combined, concentrated to 2.2 mg/ml, aliquotted, snap frozen in liquid nitrogen and stored at −80 °C.

## Sample preparation for cryo-EM

For preparation of cryo-EM grids, purified CAK (CDK7 D97N) was diluted five times in cryo-EM buffer (20 mM HEPES-KOH pH 7.9, 200 mM KCl, 2 mM MgCl$_2$) and supplemented with 2 mM ATPγS (Sigma), 50 µM Samuraciclib (MedChemExpress), or 40 µM THZ1 (Merck Millipore). Samples were then incubated for either 5 min (ATPγS and Samuraciclib) or 30 min (THZ1) at RT. Cryo-EM grids were prepared by applying 4 µl sample to UltrAuFoil R1.2/1.3 holey gold grids (Quantifoil Microtools) that had been plasma cleaned for 50 s using a Tergeo plasma cleaner (PIE Scientific). Grids were blotted for 1, 1.5, 1.5, or 2 s at 5 °C and 100% humidity using a Vitrobot Mark IV (Thermo Fisher Scientific) before plunge-freezing in liquid ethane at liquid nitrogen temperature. Grids were then clipped into autogrid cartridges.

## Cryo-EM data collection, data processing, and model building

Data were collected on cryo-transmission electron microscopes operated at 300 kV, either at the London Cryo-EM Consortium (LonCEM) facility at The Francis Crick Institute, or at the electron Bio-Imaging Centre (eBIC) at Diamond Light Source, Oxfordshire. For the CAK (CDK7 D97N)-ATPγS complex, data were collected on a Krios microscope (Thermo Fisher Scientific) equipped with a K3 BioQuantum detector and Gatan imaging filter (GIF) at LonCEM. Imaging was performed at a nominal magnification of 165,000x, giving a pixel size of 0.52 Å/pixel, and a total dose of 63 e$^-$/Å$^2$, yielding 17,516 movies in TIFF format from one grid. For the CAK (CDK7 D97N)-Samuraciclib complex, data were collected on a Krios equipped with a K3 BioQuantum detector and GIF at eBIC. Imaging was performed at a nominal magnification of

×165,000, giving a pixel size of 0.513 Å/pixel, and a total dose of 70 e⁻/Å², yielding 10,678 movies in TIFF format from one grid. For the CAK (CDK7 D97N)-THZ1 complex, data were collected on a Krios equipped with a Falcon 4i detector (Thermo Fisher Scientific) and Selectris X energy filter (Thermo Fisher Scientific) at eBIC. Imaging was performed at a nominal magnification of ×215,000, giving a pixel size of 0.576 Å/pixel, and a total dose of 70 e⁻/Å², yielding 9043 movies in EER format from one grid.

All three datasets followed the same general data processing strategy. Data were initially processed in cryoSPARC live (Punjani et al, 2017), during which movies were binned 2× during motion correction. In cryoSPARC live, particles were blob-picked using a blob diameter of 80–110 Å and extracted with a box size of 160 × 160 pixels, then classified into 50 2D classes using default parameters. Two additional rounds of particle picking were performed in the main cryoSPARC interface, using either selected classes from the live processing or projections from a three-dimensional (3D) reconstruction as templates, to maximise particle numbers and rare orientations. Particles from each of the three picking strategies were extracted from accepted micrographs using a box size of 160 × 160 pixels (either with or without downscaling to 80 × 80 pixels), then classified into 200 2D classes. Selected classes from the three classifications were combined and duplicate particles were removed, before re-extraction with a box size of 160 × 160 pixels. An additional duplicates removal step was performed before the final particle set was exported and converted to a star file format compatible with RELION using the csparc2star.py program within the PYEM package (Asarnow et al (2019)).

In RELION 5.0 (Scheres, 2012), movies were motion corrected while binning 2×, such that exported particles from cryoSPARC could be extracted from motion corrected movies using a box size of 160 × 160 pixels. Extracted particles were subjected to a consensus 3D refinement before being classified into four classes by alignment-free 3D classification ($\tau = 16$, mask diameter 104 Å) for selection of the highest-quality particle subset. This was subjected to CTF refinement and Bayesian polishing to further improve the map quality and resolution before a final refinement, which was followed by post-processing using a tight mask to generate the final reconstruction. For the Samuraciclib and THZ1 datasets, the box size was expanded to 180 × 180 pixels (without re-scaling) during Bayesian polishing to account for delocalised signal. Additionally, the final refinement for the Samuraciclib dataset was performed using Blush regularisation (Kimanius et al, 2024), which resulted in improved ligand density. For the ATPγS dataset, the processing pipeline was performed using a raw pixel size of 0.52 Å, which was then adjusted to the calibrated raw pixel size of 0.51 Å (binned pixel size of 1.02 Å) during the final post-processing job.

For all three complexes, initial models were generated by rigid-body fitting of existing high-resolution cryo-EM structures of WT CAK-ligand complexes (PDBs 8P6Y, 8P6V, and 8ORM) (Cushing et al, 2024) into the relevant reconstructions in UCSF ChimeraX (Goddard et al, 2018; Pettersen et al, 2021). These were then edited by iterative rounds of manual model building in Coot (Emsley et al, 2010) and real-space refinement in PHENIX (Afonine et al, 2018). Water molecules were manually modelled in Coot where their presence was supported by the density, the existence of nearby hydrogen bonding partner(s), and a Q-score (Pintilie et al, 2020) of 0.70 or higher. Refinement statistics are provided in Table EV1.

## Expression and purification of CDK2

CDK2 was expressed in *Escherichia coli* Rosetta (DE3) pLysS cells. A 1 litre culture containing 100 μg/ml ampicillin and 34 μg/ml chloramphenicol was inoculated to OD = 0.1 with a pre-culture that had been grown overnight at 37 °C. Cultures were incubated at 30 °C until reaching OD = 0.8. Protein expression was induced by the addition of 1 mM IPTG. Subsequently, the cultures were incubated at 25 °C for approx. 20 h, harvested, frozen in liquid nitrogen and stored at −80 °C until further use.

For protein purification, bacterial pellets were resuspended in lysis buffer supplemented with protease inhibitors and DNase I. The cells were lysed by sonication, and the lysate was clarified by spinning at 18,000 rpm (approx. 19,000 × g) for 30 min at 4 °C in a F0650 rotor (Beckman Coulter). The cleared lysate was incubated with pre-equilibrated Ni-NTA Superflow beads (Qiagen) for 30 min at 4 °C. The beads were washed with CDK2 purification buffer (50 mM HEPES pH 7.5, 180 mM NaCl, 5% (v/v) glycerol, 2 mM MgCl₂, 2 mM DTT) supplemented with 25 mM imidazole and the protein was eluted by raising the imidazole concentration to 300 mM. The sample buffer was exchanged using a HiPrep 26/10 desalting column (Cytiva) equilibrated with CDK2 gel filtration buffer (25 mM HEPES pH 7.9, 200 mM KCl, 5% (v/v) glycerol, 2 mM MgCl₂, 5 mM β-mercaptoethanol) to remove the high imidazole concentration. The buffer-exchanged protein sample was then incubated with TEV protease (at a protease:protein ratio of 1:10 by mass) for 2 h at RT to remove the His₆-tag. After cleavage, the sample was incubated with equilibrated Ni-NTA Superflow beads for 30 min at RT to remove the His₆-tagged protease. The flow-through was subjected to size exclusion chromatography using a Superdex 200 Increase 10/300 column. Eluted peak fractions were combined, concentrated, aliquotted, and snap frozen in liquid nitrogen for storage at −80 °C.

## In vitro kinase assays

Activity of CAK-D97N towards CDK2 was assessed by in vitro kinase reactions and western blot detection of the resulting phosphorylated product. 100 nM CAK (containing WT or D97N mutant CDK7) was incubated with 10 μM CDK2 and 2 mM ATP in a total volume of 100 μl in assay buffer (25 mM HEPES-KOH pH 7.9, 150 mM KCl, 5 mM MgCl₂, 2.5% glycerol, 5 mM β-mercaptoethanol) for 1 h at RT. Samples were taken at 0, 0.5, and 1 h. Samples were diluted 1:20 in assay buffer and quenched by mixing with SDS-PAGE loading dye. In all, 10 μl of the diluted samples were separated on NuPAGE 4–12% Bis-Tris gels, such that $N = 3$ technical replicates for a single CAK complex were run on a single gel, followed by Western transfer to nitrocellulose membranes (using a Trans-Blot Turbo system, BioRad) for immunodetection. After the transfer step, the nitrocellulose membranes were stained using Ponceau S (Thermo Fisher Scientific), and the resulting loading control was imaged using a ChemiDoc imaging system (BioRad). Subsequently, the membranes were destained by washing in PBST (1× phosphate-buffered saline supplemented with 0.1% v/v Tween 20) and immunoblotting was performed using a rabbit anti-phospho-CDK2 (Thr160) primary antibody (Cell Signaling Technology, cat. #2561) diluted 1:1000. Band intensities were quantified using ImageJ (Schneider et al, 2012).

## Statistical analysis

Statistical analyses were performed using GraphPad Prism v10. Pairwise comparisons between treatment groups and vehicle were carried out using repeated measures one-way analysis of variance (ANOVA) followed by Fisher's least significant difference post hoc test. For the analysis of P-CDK2 levels in kinase reactions using recombinant CAK-D97N or CAK-WT with recombinant CDK2 as substrate, two-way ANOVA with correction for multiple comparisons was applied.

# Data availability

Requests for further information, resources, and reagents should be directed to and will be fulfilled by the lead contact, SA (simak.ali@imperial.ac.uk). Plasmids and cell lines generated in this study can be obtained from the lead contact, SA (simak.ali@imperial.ac.uk). The RNA-seq data are publicly available through the NCBI GEO repository using the accession number GSE279730. The cryo-EM maps and atomic coordinates for the CAK (CDK7-D97N)-ATPγS, CAK (CDK7-D97N)-Samuraciclib, and CAK (CDK7-D97N)-THZ1 structures have been deposited to the EM Data Resource with accession codes EMD-52206, EMD-52207, and EMD-52205 and to the Protein Data Bank (PDB) with accession codes PDB-9HIY, PDB-9HJ0, and PDB-9HIX, respectively. This paper does not report original code. Any additional information required to reanalyse the data reported in this paper is available from the lead contacts upon request.

The source data of this paper are collected in the following database record: biostudies:S-SCDT-10_1038-S44318-025-00554-6.

# Peer review information

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

## Acknowledgements

We thank S. Omarjee, J. Carroll and A. Bahl for their generous gift of cell lines and samuraciclib, respectively. We thank C. Richardson for support with high-performance computing; R. Knight for support of insect cell culture; and K. Kondas, M. Stubbs, S. Hearnshaw and R. van Montfort for help with biophysical assays. SA, CLB, LB and C-FL received funding for this work from Prostate Cancer UK (grant number RIA19-ST2-00). BJG was supported by a career development fellowship from the Medical Research Council of the UK (grant number MR/V009354/1). VIC was funded by an ICR PhD studentship. EO was funded by the Imperial MRC MultiSci doctoral training partnership PhD programme, with industrial funding from Carrick Therapeutics. We acknowledge Diamond for access and support of the cryo-EM facilities at the UK national electron Bio-Imaging Centre (eBIC), proposal BI33974, and we acknowledge data collection at the London Consortium for High-Resolution Cryo-EM (LonCEM) facility, supported by Wellcome Grant No. 206175/Z/17/Z and partner institutes. We thank eBIC local contacts Y. Chaban, P. Majumder, and D. Clare, as well as LonCEM manager N. B. Cronin, for support of data collection. Additional support was provided by the Imperial Experimental Cancer Medicine Centre and the Imperial NIHR Biomedical Research Centre. The views expressed are those of the authors and not necessarily those of the NHS, the NIHR or the Department of Health.

## Author contributions

**Chun-Fui Lai**: Data curation; Formal analysis; Investigation; Visualisation; Methodology; Writing—review and editing. **Victoria I Cushing**: Data curation; Investigation; Visualisation; Methodology. **Ellen Olden**: Investigation. **Charlotte L Bevan**: Supervision; Funding acquisition; Writing—review and editing. **R Charles Coombes**: Supervision; Project administration; Writing—review and editing. **Basil J Greber**: Conceptualisation; Data curation; Supervision; Writing—original draft; Project administration. **Laki Buluwela**: Conceptualisation; Supervision; Funding acquisition; Writing—review and editing. **Simak Ali**: Conceptualisation; Data curation; Formal analysis; Funding acquisition; Writing—original draft; Project administration; Writing—review and editing.

Source data underlying figure panels in this paper may have individual authorship assigned. Where available, figure panel/source data authorship is listed in the following database record: biostudies:S-SCDT-10_1038-S44318-025-00554-6.

## Disclosure and competing interests statement

SA and RCC are named inventors on patents concerning CDK7 inhibitors, which have been licensed to Carrick Therapeutics, own shares in and has received royalties from Carrick Therapeutics. RCC is on the advisory board for Carrick Therapeutics. SA, LB and CLB have also received research funding from Carrick Therapeutics. EO was funded by a PhD programme with funding from Carrick Therapeutics.

# Expanded View Figures

**Figure EV1. Samuraciclib-resistant 22Rv1 cells are cross-resistant to other ATP-competitive CDK7 inhibitors.**

(A) Mean $IC_{50}$ values from $n = 3$ independent experiments for 22Rv1 and 22Rv1-SamR cells. These results are summarised in Fig. 1. Error bars = SEM. (B) Densitometric quantification of immunoblots from $n = 3$ independent experiments in which 22Rv1 and 22Rv1-SamR cells were treated with the indicated concentrations of Samuraciclib or SY1365. Signal intensities are shown relative to the appropriate vehicle (0 nM) controls. One of the immunoblots used for the quantification is shown in Fig. 1D. Error bars show SEM. Pairwise comparisons between treated groups and vehicle were performed using repeated measures of one-way ANOVA followed by Fisher's LSD post hoc test (uncorrected). Asterisks indicate significance (*$P < 0.05$, **$P < 0.01$). $P$ value for P-Ser5, 22Rv1 cells: Veh vs Sam (100 nM) $P = 0.0093$; Veh vs Sam (1000 nM) $P = 0.0232$. 22Rv1-SamR cells: Veh vs SY1365 (5 nM) $P = 0.0227$; Veh vs SY1365 (50 nM) $P = 0.0052$. For P-CDK2, 22Rv1 cells: Veh vs Sam (100 nM) $P = 0.0148$; Veh vs Sam (1000 nM) $P = 0.0017$; Veh vs SY1365 (50 nM) $P = 0.0268$. 22Rv1-SamR cells: Veh vs Sam (1000 nM) $P = 0.0027$; Veh vs SY1365 (50 nM) $P = 0.0102$. For β-actin, 22Rv1-SamR cells: Veh vs sam (1000 nM) $P = 0.0441$. No asterisk indicates a non-significant difference ($P > 0.05$). (C) RNA-seq was performed using six RNA samples prepared from 22Rv1 and 22Rv1-SamR cells. Shown are normalised read counts for CDK7 and its interacting partners cyclin H (CCNH) and MAT1 (MNAT1). Circles represent the normalised read counts for each of the six biological replicate RNA samples. (D, E) Genome browser snapshots (hg38) of RNA-seq data for 22Rv1 (SAMSEN) and 22Rv1-SamR (SAMRES) cells are shown. The vertical bars show positions of two known CDK7 SNPs (rs2972388 (exon 2) and rs34584424 (exon 10)), which are present in the Sam-sensitive and resistant cells at a ratio of 1:1, suggestive of the presence of 2 CDK7 alleles in 22Rv1 cells. Also evident is a single nucleotide difference (exon 5; c.289G>A) in the codon encoding aspartate 97 (p.Asp97Asn), which is seen only in 22Rv1-SamR cells. Source data are available online for this figure.

## A

| | | IC50 (nM) | | 95% CI | |
|---|---|---|---|---|---|
| | | 22Rv1 | 22Rv1-Sam | 22Rv1 | 22Rv1-Sam |
| **Non-covalent** | Samuraciclib | 107.1 | 1719 | 93.3 to 123.1 | 1473 to 2005 |
| | LDC4297 | 38.9 | 1717 | 32.9 to 45.0 | 1503 to 1962 |
| | SY-5609 | 3.7 | 5136 | 3.0 to 4.7 | 4326 to 6124 |
| **Covalent** | SY-1365 | 2.4 | 2.4 | 2.1 to 2.7 | 1.8 to 3.3 |
| | THZ1 | 21.4 | 13.9 | 19.4 to 23.9 | 12.2 to 15.8 |
| | YKL-5-124 | 6.2 | 12.4 | 5.4 to 7.1 | 10.5 to 14.6 |

## B

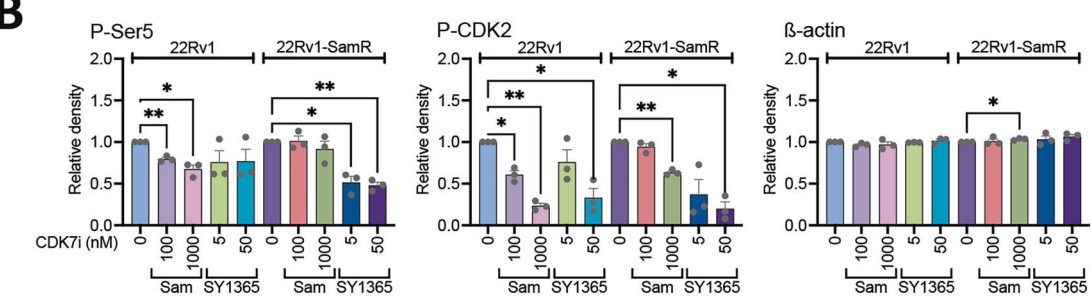

## C

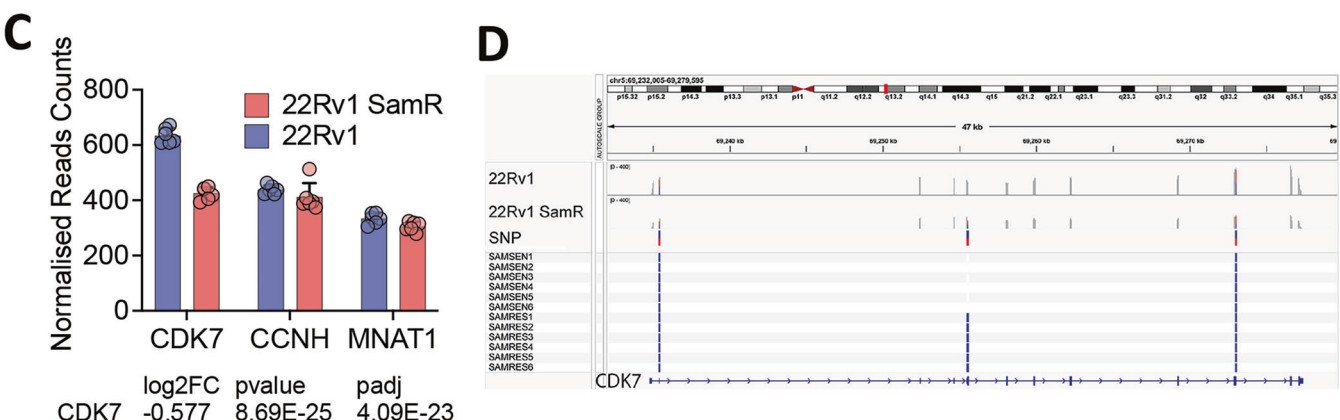

## D

## E

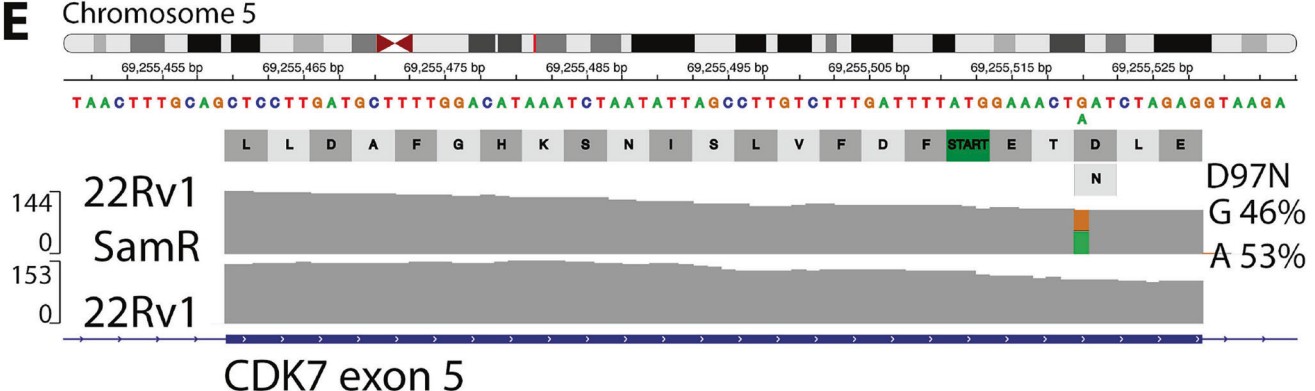

Chromosome 5

CDK7 exon 5

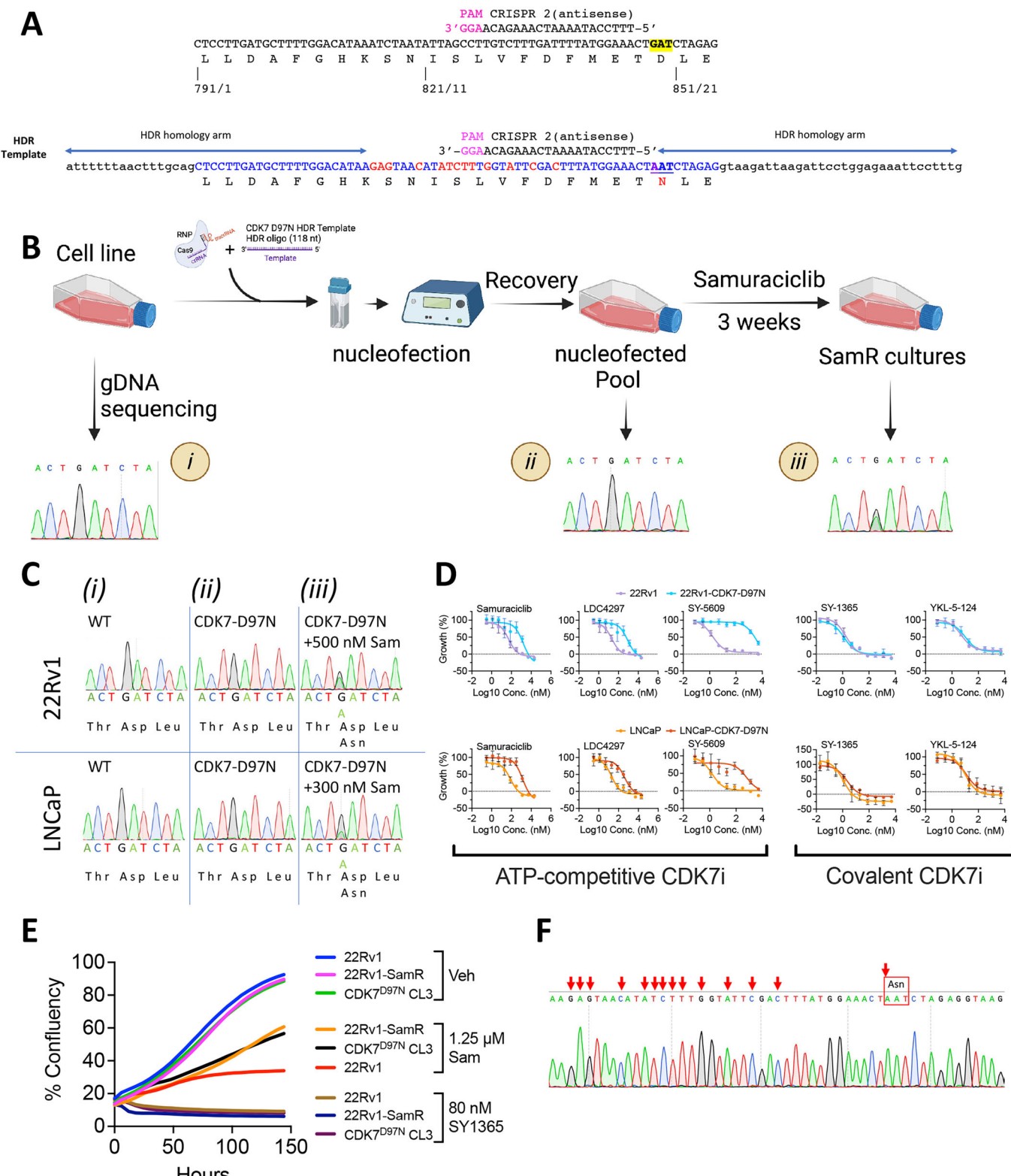

**A**

PAM CRISPR 2(antisense)
3'GGAACAGAAACTAAAATACCTTT-5'
CTCCTTGATGCTTTTGGACATAAATCTAATATTAGCCTTGTCTTTGATTTTATGGAAACT**GAT**CTAGAG
L L D A F G H K S N I S L V F D F M E T D L E
|           |           |
791/1       821/11      851/21

HDR
Template

HDR homology arm                                                    HDR homology arm

PAM CRISPR 2(antisense)
3'-GGAACAGAAACTAAAATACCTTT-5'
attttttaactttgcagCTCCTTGATGCTTTTGGACATAAGAGTAACATATCTTTGGTATTCGACTTTATGGAAACT**AAT**CTAGAGgtaagattaagattcctggagaaattcctttg
L L D A F G H K S N I S L V F D F M E T N L E

**B**

Cell line

RNP / Cas9 + CDK7 D97N HDR Template / HDR oligo (118 nt)

nucleofection → Recovery → nucleofected Pool → Samuraciclib 3 weeks → SamR cultures

gDNA sequencing
*i* ACTGATCTA
*ii* ACTGATCTA
*iii* ACTGATCTA

**C**

*(i)* WT / *(ii)* CDK7-D97N / *(iii)* CDK7-D97N +500 nM Sam

22Rv1:
(i) ACTGATCTA — Thr Asp Leu
(ii) ACTGATCTA — Thr Asp Leu
(iii) ACTGATCTA — Thr Asp Leu / Asn

LNCaP:
WT / CDK7-D97N / CDK7-D97N +300 nM Sam
(i) ACTGATCTA — Thr Asp Leu
(ii) ACTGATCTA — Thr Asp Leu
(iii) ACTGATCTA — Thr Asp Leu / Asn

**D**

Samuraciclib / LDC4297 / SY-5609 — 22Rv1, 22Rv1-CDK7-D97N
SY-1365 / YKL-5-124
Growth (%) vs Log10 Conc. (nM)

Samuraciclib / LDC4297 / SY-5609 — LNCaP, LNCaP-CDK7-D97N
SY-1365 / YKL-5-124

ATP-competitive CDK7i                    Covalent CDK7i

**E**

% Confluency vs Hours

22Rv1 / 22Rv1-SamR / CDK7^D97N CL3 — Veh
22Rv1-SamR / CDK7^D97N CL3 / 22Rv1 — 1.25 µM Sam
22Rv1 / 22Rv1-SamR / CDK7^D97N CL3 — 80 nM SY1365

**F**

AA G AG T AA C A T A T C T T T G G T A T T C G A C T T T A T G G AA A C T AA T C T A G A G G T AA G
Asn

◀  **Figure EV2.   Development of CDK7-D97N mutation in cancer cell lines using CRISPR/Cas9 mutagenesis.**

(A, B) Shown are the CRISPR sequences and the sequence of the homology-directed recombination (HDR) template for introduction of the CDK7-D97N mutation. The sequences in blue show the CDK7 exon 5 sequences, and the sequences in red represent silent changes introduced in the template for facilitating screening for the mutation. Underlined is the codon encoding asparagine at position 97 to create the p.Asp97Asn (D97N) mutation in the CDK7 gene. (B) The strategy for enrichment of cells encoding CDK7-D97N. (C) Sequencing chromatograms for the region around Asp97 in unmutated cells (i) in gDNA prepared from 22Rv1 and LNCaP cells. Sequencing chromatograms for cells following nucleofection and culturing in the absence (ii) or the presence (iii) of Sam. (D) Growth inhibition for increasing concentrations of the indicated drugs was carried out using Sam-selected D97N knock-in cells (22Rv1-CDK7-D97N, LNCaP-CDK7-D97N (iii)) alongside parental Sam-sensitive cells; means and SEM from $n = 2$ independent experiments are shown. (E) The cell lines were cultured in the presence of the indicated drugs, with confluency being measured every 6 h using an Incucyte Zoom. Non-linear regression analysis was used for curve fitting. The results of $n = 3$ independent experiments are summarised, with error bars omitted for ease of viewing. Calculated doubling times with 95% confidence intervals (h) for vehicle-treated cells: 22Rv1 (49.4; 45.3–54.1), 22Rv1-SamR (45.4; 42.2–49.0), 22Rv1 homozygous CDK7-D97N CL3 (48.8; 45.1–53.0). (F) Sanger sequencing chromatogram for gDNA prepared from the 22Rv1 CDK7-D97N CL3. Source data are available online for this figure.

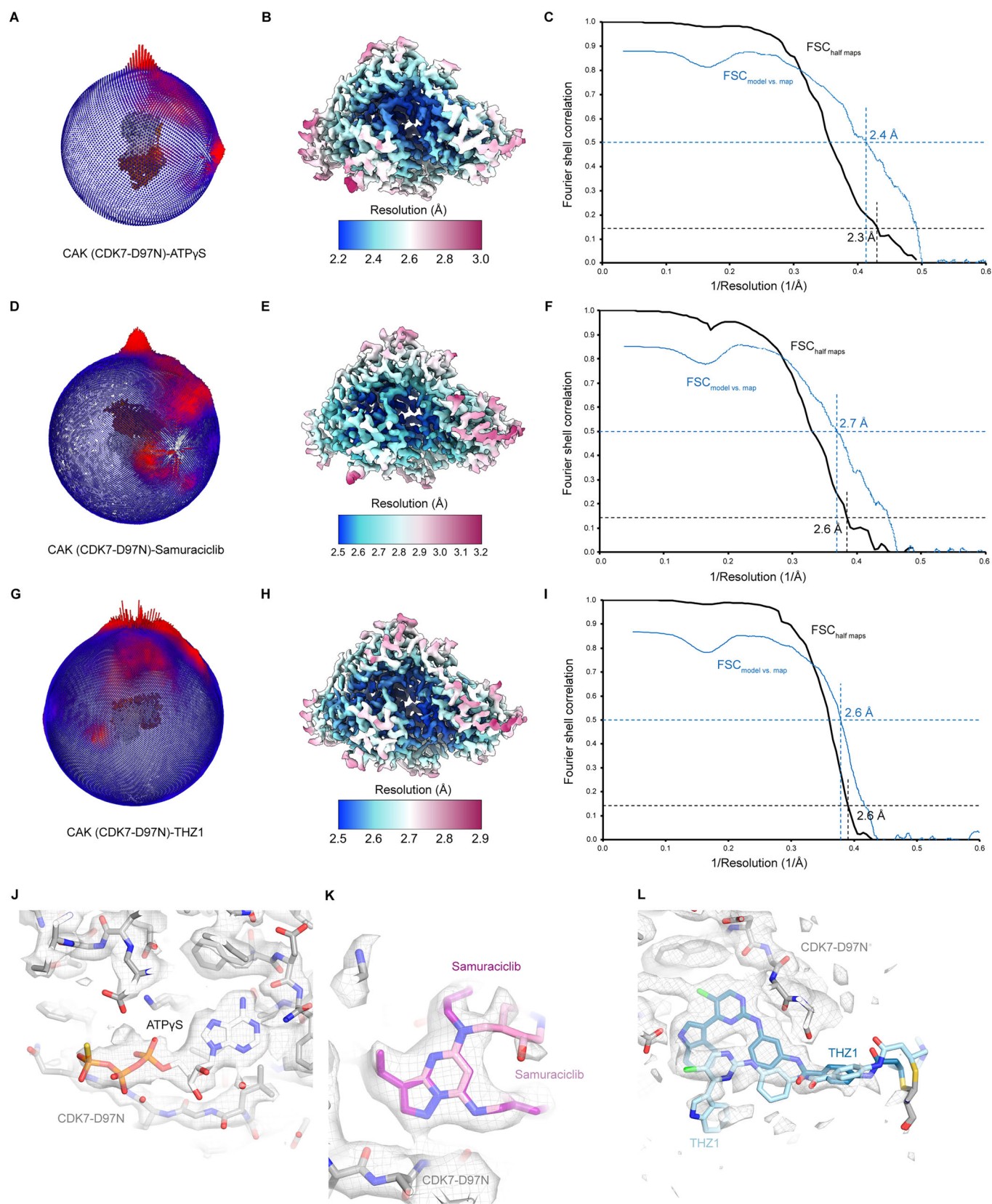

**Figure EV3. Validation measures for the cryo-EM reconstructions and additional ligand views.**

(A) Plot of the orientation distribution of the particle images entering the CAK D97N-ATPγS cryo-EM reconstruction. (B) Local resolution estimation for the CAK D97N-ATPγS cryo-EM reconstruction. (C) FSC plots to assess the overall resolution of the CAK D97N-ATPγS cryo-EM reconstruction (black line) and the model-to-map fit of the refined coordinate model (blue line). FSC thresholds of FSC = 0.143 and FSC = 0.5 were used for half-map and model-to-map comparisons, respectively (Rosenthal and Henderson, 2003). (D–F) As (A–C), but for the CAK D97N-Samuraciclib cryo-EM reconstruction. (G–I) As (A–C), but for the CAK D97N-THZ1 cryo-EM reconstruction. (J–L) Additional views of the ligands in the cryo-EM density. Source data are available online for this figure.

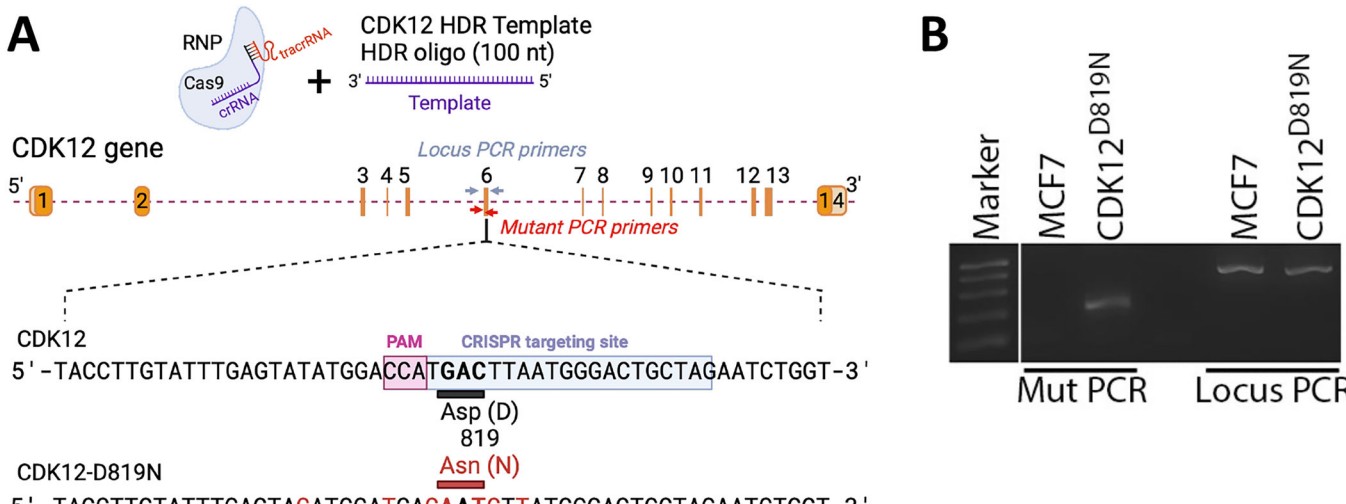

**Figure EV4.  CDK12-D819N CRISPR-Cas9 knock-in in MCF7 cells.**

(A) Strategy for generating the D819N mutation in the CDK12 gene. The full sequence of the donor template is: 5′-TTCGTCTTTATGTAGGTGCCTTTTACCTTG-TATTTGAGTACATGGATCACAATCTTATGGGACTGCTAGAATCTGGTTTGGTGCACTTTTCTGAGGACCA-3′. (B) PCR was carried out using gDNA and primers amplifying the region around exon 6 ("locus PCR"; 5′-CGCCCAGCCACAGAAGATTA-3′ and 5′-GAGGAGAAGAGGGAAAGTGCTTAA-3′, product size: 458 bp). PCR using primers, one of which is located within the region containing base changes incorporated in the donor template ("mutant PCR"; 5′-GGACTTGAGGCATTGTTATTT-3′ and 5′-CCATAAGATTGTGATCCATG-3′, product size: 116 bp), was carried out with gDNA prepared from cells following RNP CRISPR knock-in but prior to selection with (R)-CR8.

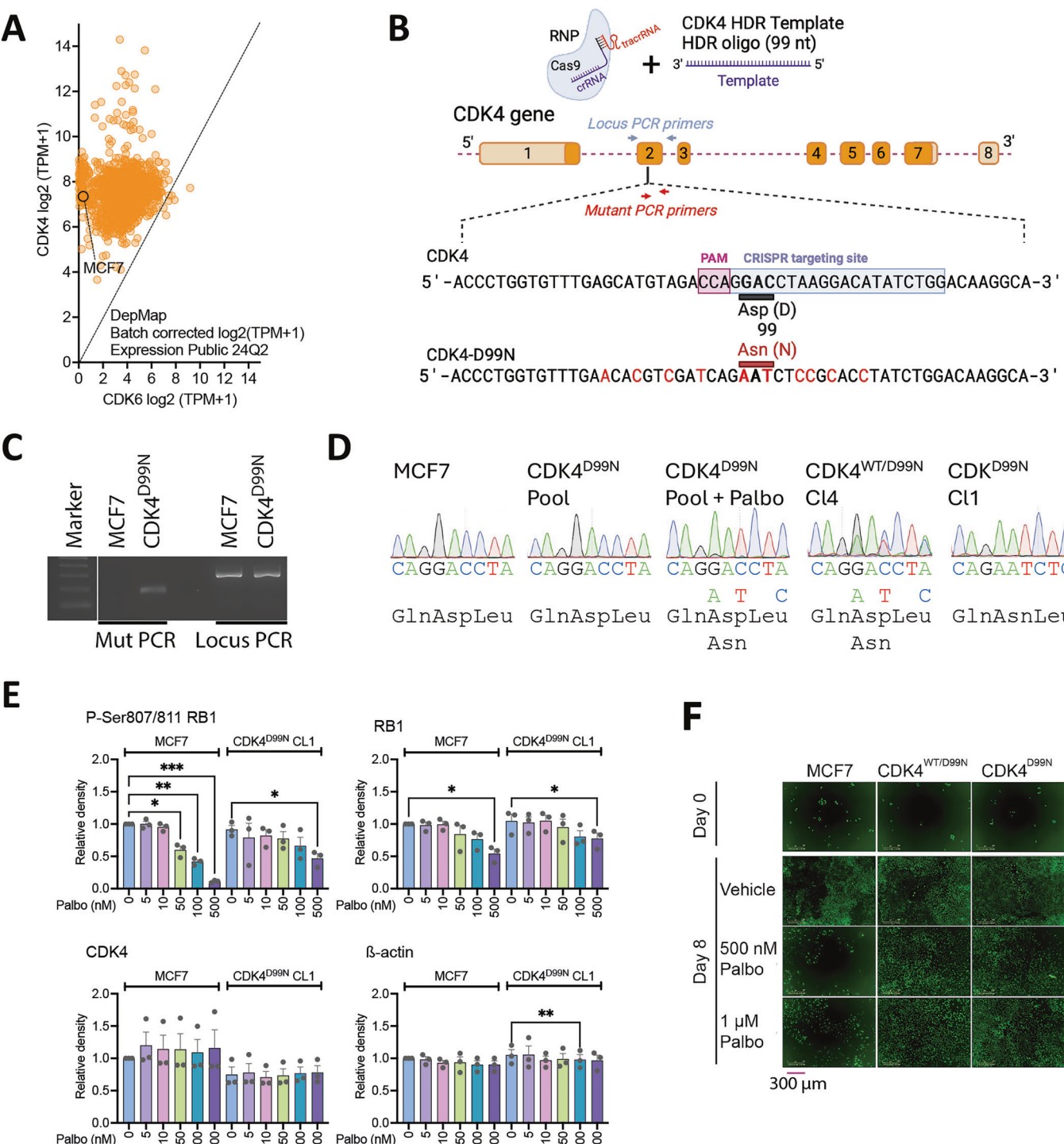

◀  **Figure EV5.   CDK4-D99N CRISPR-Cas9 knock-in in MCF7 cells.**

(A) Batch corrected expression public 24Q2 data was acquired for CDK4 and CDK6 from the DepMap portal. (B) Strategy for generating the D99N mutation in the CDK4 gene. The full sequence of the donor template is: 5′-CCCGAACTGACCGGGAGATCAAGGTAACCCTGGTGTTTGAACACGTCGATCAGAATCTCCGCACCTATCTGGA-CAAGGCACCCCCACCAGGCTTGCCAGCCGAAA-3′. (C) Knock-ins were confirmed by PCR of gDNA using mutant-specific primers having the sequences, 5′-ACACGTCGATCAGAATCTCC-3′ and 5′-ATCCACCTCTCAATGCCTAC-3′. Locus-specific primers used for gDNA PCR had the sequences 5′-AGGTGGGGTGTGAT-GATCTG-3′, 5′AAGGGGAGGTACAGATGCAC-3′. (D) Sanger sequencing of DNA generated using locus PCR primers. Sequencing was performed using gDNA prepared from MCF7 cells after nucleofection (CDK4-D99N Pool), gDNA prepared from the CDK4-D99N Pool after culturing in the presence of 1 μM Palbociclib for 9 passages over a period of 3 months. Individual clones were isolated from the Palbociclib-selected pools; sequences of heterozygous and homozygous mutant clones are shown. (E) ImageJ was used to quantify the immunoblotting results for $n = 3$ independent experiments in which MCF7 cells and MCF7-CDK4-D99N CL1 cells were treated with Palbociclib for 48 h. Error bars = SEM. Pairwise comparisons between treated groups and vehicle were performed using repeated measures of one-way ANOVA followed by Fisher's LSD post hoc test (uncorrected). For P-Ser807/811 RB1 MCF7 cells: Veh vs Palbo (50 nM) $P = 0.0213$; Veh vs Palbo (100 nM) $P = 0.0026$; Veh vs Palbo (500 nM) $P = 0.0001$. CDK4$^{D99N}$ CL1 cells: Veh vs Palbo (500 nM) $P = 0.0302$. For RB1, MCF7 cells: Veh vs Palbo (500 nM) $P = 0.0236$. Veh vs Palbo (500 nM) $P = 0.0143$. For β-actin, CDK4$^{D99N}$ CL1 cells: Veh vs Palbo (100 nM) $P = 0.0063$. No asterisk indicates a non-significant difference ($P > 0.05$). (F) Representative images of cultures 24 h after seeding, when Palbociclib was added (day 0) and 8 days following the start of treatment. Source data are available online for this figure.

