## [Peer Review File · The EMBO Journal]

Resistance to CDK7 inhibitors directed by acquired mutation of a conserved residue in cancer cells

Chun-Fui Lai, Victoria Cushing, Ellen Olden, Charlotte Bevan, R. Coombes, Basil Greber, Laki Buluwela, and Simak Ali

Corresponding authors: Simak Ali (simak.ali@imperial.ac.uk) , Laki Buluwela (l.buluwela@imperial.ac.uk), Basil Greber (basil.greber@icr.ac.uk)

Review Timeline:

Submission Date:	17th Dec 24
Editorial Decision:	4th Mar 25
Revision Received:	17th Jun 25
Editorial Decision:	22nd Jul 25
Revision Received:	28th Jul 25
Accepted:	13th Aug 25

Editor: Daniel Klimmeck

Transaction Report:

Dear Dr Ali,

Thank you again for the submission of your manuscript (EMBOJ-2024-119953) to The EMBO Journal. Please accept my sincere apologies for getting back to you with this unusual delay, which is due to protracted referee input at this time as well as detailed discussion on your work in the editorial team. As mentioned earlier, your study was assessed by three reviewers with expertise in cell cycle control, cancer CDK biology and related drug development, whose comments are enclosed below.

As you will see from the experts' reports, the referees acknowledge the analysis and potential interest and value of your findings. However, they also express important issues regarding the completeness of your study and physiological relevance of the results, which need to be addressed thoroughly to make them supportive of publication in the EMBO Journal. Further, the reviewers raise a number of issues related to the presentation of the findings, additional controls and improved methods annotation required, statistics applied and overall discussion of related literature, that would need to be conclusively addressed to achieve the level of robustness and clarity needed for The EMBO Journal.

Given the overall interest stated and broader angle of your findings, we are able to invite you to revise your manuscript experimentally to address the referees' comments. I need to stress though that we do require strong support from the referees on a revised version of the study in order to move on to publication of the work.

Please feel free to contact me if you have any questions or need further input on the referee comments.

When submitting your revised manuscript, please carefully review the instructions below.

Please feel free to approach me any time should you have additional questions related to this.

Thank you for the opportunity to consider your work for publication.

I look forward to your revision.

Best regards,

Daniel Klimmeck

Daniel Klimmeck, PhD
Senior Editor
The EMBO Journal

Instruction for the preparation of your revised manuscript:

- 1) a .docx formatted version of the manuscript text (including legends for main figures, EV figures and tables). Please make sure that the changes are highlighted to be clearly visible.
- 2) individual production quality figure files as .eps, .tif, .jpg (one file per figure).
- 3) a .docx formatted letter INCLUDING the reviewers' reports and your detailed point-by-point response to their comments. As part of the EMBO Press transparent editorial process, the point-by-point response is part of the Review Process File (RPF), which will be published alongside your paper.
- 4) a complete author checklist, which you can download from our author guidelines ([https://wol-prod-cdn.literatumonline.com/pb-assets/embo-site/Author Checklist%20-%20EMBO%20J-1561436015657.xlsx](https://wol-prod-cdn.literatumonline.com/pb-assets/embo-site/Author%20Checklist%20-%20EMBO%20J-1561436015657.xlsx)). Please insert information in the checklist that is also reflected in the manuscript. The completed author checklist will also be part of the RPF.
- 5) Please note that all corresponding authors are required to supply an ORCID ID for their name upon submission of a revised

manuscript.

6) It is mandatory to include a 'Data Availability' section after the Materials and Methods. Before submitting your revision, primary datasets produced in this study need to be deposited in an appropriate public database, and the accession numbers and database listed under 'Data Availability'. Please remember to provide a reviewer password if the datasets are not yet public (see <https://www.embopress.org/page/journal/14602075/authorguide#datadeposition>).

7) Our journal encourages inclusion of *data citations in the reference list* to directly cite datasets that were re-used and obtained from public databases. Data citations in the article text are distinct from normal bibliographical citations and should directly link to the database records from which the data can be accessed. In the main text, data citations are formatted as follows: "Data ref: Smith et al, 2001" or "Data ref: NCBI Sequence Read Archive PRJNA342805, 2017". In the Reference list, data citations must be labeled with "[DATASET]". A data reference must provide the database name, accession number/identifiers and a resolvable link to the landing page from which the data can be accessed at the end of the reference. Further instructions are available at .

8) At EMBO Press we ask authors to provide source data for the main and EV figures. Our source data coordinator will contact you to discuss which figure panels we would need source data for and will also provide you with helpful tips on how to upload and organize the files.

Numerical data can be provided as individual .xls or .csv files (including a tab describing the data). For 'blots' or microscopy, uncropped images should be submitted (using a zip archive or a single pdf per main figure if multiple images need to be supplied for one panel). Additional information on source data and instruction on how to label the files are available at .

9) We replaced Supplementary Information with Expanded View (EV) Figures and Tables that are collapsible/expandable online (see examples in <https://www.embopress.org/doi/10.15252/embj.201695874>). A maximum of 5 EV Figures can be typeset. EV Figures should be cited as 'Figure EV1, Figure EV2' etc. in the text and their respective legends should be included in the main text after the legends of regular figures.

11) For data quantification: please specify the name of the statistical test used to generate error bars and P values, the number (n) of independent experiments (specify technical or biological replicates) underlying each data point and the test used to calculate p-values in each figure legend. The figure legends should contain a basic description of n, P and the test applied. Graphs must include a description of the bars and the error bars (s.d., s.e.m.).

We realize that it is difficult to revise to a specific deadline. In the interest of protecting the conceptual advance provided by the work, we recommend a revision within 3 months (2nd Jun 2025). Please discuss the revision progress ahead of this time with the editor if you require more time to complete the revisions.

Referee #1:

Lai et al., generated prostate cancer cells resistant to Samuraciclib and identified a D97N mutation in CDK7 that reduces binding affinity to ATP-competitive CDK7i while preserving sensitivity to covalent inhibitors. Cryo-EM analysis revealed subtle conformational changes affecting inhibitor binding. Cyclin-dependent kinases regulate both the cell cycle and transcription, making them a key cancer therapy target. However, resistance to inhibitors targeting these kinases is a fundamental issue in oncology. Similar mutations in CDK4, CDK6, and CDK12 confer resistance to their respective inhibitors, highlighting a conserved resistance mechanism and the importance of the findings shown here.

The manuscript is well written, easy to follow and describes a spontaneously occurring mutation that has importance across both the cell cycle and cancer fields. However, I have minor concerns that need to be addressed:

1.) The CDK7 (D97N) mutation occurs monoallelically, but the authors do not address why biallelic mutations were not observed. Did they isolate clones or assess only pooled cells? Could both allelic states be present within the population? Similarly, why did CRISPR editing yield only heterozygous modifications? The authors successfully generated homozygous CDK4 (D99N) clones in MCF7, so why was this not the case for CDK7/CDK12?

2.) In Figure 5B, the authors show enrichment of the CDK12 (D819N) mutation in MCF7 cells following CRISPR knock-in and CR8 treatment. However, the chromatogram suggests the presence of additional mutations near this site. How can the authors conclude that the observed phenotype is solely due to D819N?

3.) In Figure 1D, p-POLII levels in 22Rv1 cells do not show a clear reduction upon Samuraciclib treatment, despite the authors' claims. In contrast, Figure 2E demonstrates a more expected response in MCF7.

4.) In Figure 6C, the authors do not specify whether CDK4 (D99N) CL1 cells are heterozygous or homozygous. They also claim a reduction in baseline Rb phosphorylation at S807/811 compared to WT, possibly due to altered CDK4 activity. However, no apparent difference in baseline phosphorylation is visible on the blot without Palbociclib treatment.

5.) Throughout the study, the impact of these mutations on kinase activity remains unclear. Assessing kinase activity in vitro or in homozygous clones would be necessary to clarify this. Although the cryo-EM structure suggests minimal functional impact, the authors still claim an effect on CDK4 activity without direct evidence.

Referee #2:

The paper by Lai et al., entitled "Mutation of CDK7 at a conserved residue identifies a common mechanism of acquired resistance to CDDK inhibitors in cancer" describes the characterization of a spontaneous mutation in CDK7 that was discovered in a prostate cancer cell model of resistance to a CDK7 inhibitor, Samuraciclib, which has been shown by this team to have efficacy in ER+ breast cancer patients progressing on a prior CDK4/6 inhibitor treatment. Elegantly, this mutation was introduced in one breast cancer and two prostate cancer cell lines by targeted recombination, demonstrating that the mutation is sufficient for resistance. This residue is conserved in other CDKs and mutagenesis of the corresponding residue in CDK4, CDK6, and CDK12 also resulted in resistance to inhibitors of these enzymes. Mechanistically, although cryoEM structures of CDK7 did not reveal major changes in conformation of the protein but rather a greater structural heterogeneity at the active site, the author demonstrated using a nanoLuc assay that binding of all non-covalent ATP-competitive inhibitors occurred with lower affinity, explaining their lower potency in cell lines with the knock-in mutation.

The results of this detailed molecular study are clear and convincing. They are original and have important implications for resistance to a growing class of anticancer drugs, and thus should be of interest for a wide readership.

My only reservation is that greater detail in the description of experimental approaches and results, as outlined below, would enhance the manuscript.

- Please provide some detail about the 22Rv1 model used as the resistance model.
- Description of Fig. 1D results with more granularity, perhaps including quantitation of bands in the Western analysis, would be desirable as some of the changes are more modest than those shown in Figure 2E.

- Relative growth curves are provided but the assay used to monitor cell proliferation is not mentioned.
- Fig. 2E is not described
- While the CryoEM structure suggests that the mutation maintains kinase activity, please mention if the fitness of the cell lines bearing the mutant CDKs is affected.
- In the discussion, it would be worth comparing the clinical benefits of ATP-competitive vs covalent inhibitors, which are not affected by this mechanism of resistance.

Referee #3:

The manuscript by Lai et al. reports the selection of specific mutations in the critical, conserved D97 Asp in CDK7 upon continuous treatment with ATP-competitors. Continuous treatment of prostate cancer cells with Samuraciclib, a non-covalent ATP-competitive CDK7i, leads to the selection of resistant clones characterized by a single allele alteration in exon 5 (c.289G>A; p.Asp97Asn; D97N). This change results in resistance to multiple non-covalent, ATP-competitive CDK7i, but not to covalent inhibitors. The CDK7-D97N mutation does not induce major alterations of the CDK7 active site, consistent with the observation that the mutant enzyme might be still active. However, the affinity of the non-covalent inhibitors to CDK7-D97N is reduced. These data are reproduced in different cell types after CRISPR-mediated gene editing of this residue in CDK7.

Since the Asp residue is conserved across the CDK family, the authors repeat CRISPR-mediated gene editing of CDK12, CDK4 or CDK6 to demonstrate that a similar change to Asn reduces the binding and effect of non-covalent ATP-competitors. Similarly, the change per se does not seem to significantly alter the growth of the untreated cells, suggesting active kinases in the presence of that mutation. Overall, these data suggest that the Asp to Asn mutation in the critical Asp of the hinge domain of CDKs may have mild effects on the activity of the kinase, while decreasing the affinity for specific ATP-competitors.

The data are interesting, and the fact that specific conservative mutations in this residue may alter the binding of ATP or ATP-competitors is not unexpected. Yet, the description of Asp-to-Asn mutations that maintain the endogenous function of the protein while decreasing the response to clinically-relevant inhibitors opens the possibility of finding similar mutations in patients. Since these mutations have not been found in patients yet, it remains to be seen whether this is a reality. There are a couple of unexplored reasons why this is not frequent (or even seen) in patients even after whole-exome sequencing (in breast cancer; Wander et al., 2020 among others).

If I understand properly the data, the manuscript describes the selection of this mutation in a single clone (or pool?), 22Rv1-SamR, in Figure 1. The rest of assays are performed after CRISPR-mediated gene editing. How many times that mutation has been observed in different clones? And in different cell lines? Could it be possible that this mutation is pre-existing in 22Rv1 cells? Could it be detected in untreated 22Rv1 prostate cancer using high-depth sequencing methods?

The central Asp in the CDK hinge is a critical residue to CDK activity. Although mutant clones seem to grow similarly in the absence of inhibitors, a more direct comparison on the relative binding of ATP versus different ATP-competitors would be informative. In addition, a direct quantification of phosphorylation activity using more direct molecular quantification methods (rather than WB in cells) to compare ATP versus ATP-competitor activity would also be important to understand whether these mutations are never seen in vivo due to minor but significant defects in the endogenous kinase activity.

Related to the previous comment, the relative activity is hard to compare in WB studies as some CDK7i do not seem to be effective on RNAPol CTD sites even in parental cells (see Fig. 1D). A quantification of these signals in several repetitions would also help here.

The fact that these mutations have not been seen in patients (perhaps due to reasons discussed in the previous points) deserves some discussion. The Discussion section is very short in the present document (apart from discussing the Persky et al., 2020 study), and it would benefit from reduced sentences dedicated to repeat the results in favor of a more clinically discussion: Why these discussions have not been seen in patients? Would similar mutations affect differentially different drugs? Could eventually these mutations be targeted with complementary drugs that do not require similar binding to the same residues? Would other mutations in Asp be lethal (as this residue is typically mutated to generate kinase-dead CDKs)?

Could perhaps the title reflect that data were generated in vitro? The "common" word is also misleading here. It may apply to other CDKs but is not commonly found either in vitro or in vivo. In general, I acknowledge the value of the present study but, in the present form, it remains closer to a biochemical curiosity than a relevant description of current mechanisms of resistance in cancer.

Rebuttal Letter, Lai et al

Referee #1:

Lai et al., generated prostate cancer cells resistant to Samuraciclib and identified a D97N mutation in CDK7 that reduces binding affinity to ATP-competitive CDK7i while preserving sensitivity to covalent inhibitors. Cryo-EM analysis revealed subtle conformational changes affecting inhibitor binding. Cyclin-dependent kinases regulate both the cell cycle and transcription, making them a key cancer therapy target. However, resistance to inhibitors targeting these kinases is a fundamental issue in oncology. Similar mutations in CDK4, CDK6, and CDK12 confer resistance to their respective inhibitors, highlighting a conserved resistance mechanism and the importance of the findings shown here.

The manuscript is well written, easy to follow and describes a spontaneously occurring mutation that has importance across both the cell cycle and cancer fields. However, I have minor concerns that need to be addressed:

We are grateful for the reviewer's feedback on the work presented in our manuscript and their appreciation of the significance of our findings for the fields of cancer and cell cycle research.

1.) The CDK7 (D97N) mutation occurs monoallelically, but the authors do not address why biallelic mutations were not observed. Did they isolate clones or assess only pooled cells? Could both allelic states be present within the population? Similarly, why did CRISPR editing yield only heterozygous modifications? The authors successfully generated homozygous CDK4 (D99N) clones in MCF7, so why was this not the case for CDK7/CDK12?

From the structural and CDK7 binding studies, we would predict that the CDK7 (D97N) mutation will act in a dominant manner when the cells are treated with samuraciclib. In heterozygous cells, wild-type CDK7 would continue to be inhibited by samuraciclib (and other ATP-competitive, non-covalent CDK7i) but the D97N mutant CDK7 would not be inhibited, except at high doses. As such, a single allele mutation is most likely sufficient for selection of resistant cells and a biallelic mutation not necessary. While we did not attempt to obtain clones from the 22Rv1 SamR cultures, we would note that single allele CRISPR knockin clones are resistant to samuraciclib, supporting this premise. Notwithstanding, it is of course possible that there are cells within the 22Rv1 SamR population that have biallelic mutations. The percentages of CDK7 RNA-seq reads with G (WT) and A (mutant) base at the Asp97 codon are 46% and 53% respectively (Supplementary Figure 1D), which could be consistent with a very small proportion of the 22Rv1 SamR cells being biallelically mutant.

Regarding the knock-in cells we generated using CRISPR editing, our experience is that efficiency of knock-in mutations is much lower than that achievable for indel gene knockout approaches. We find that efficiencies for two allele knock-in are lower still.

Since a single allele knockin would be sufficient for resistance to CDK7i, we believe that most of the clones that we obtain will be single allele knock-in.

The results shown in the manuscript for CDK7 D97N knock-in cells are from pooled cells selected with Samuraciclib. We went on to isolate 10 clones from 22Rv1 CDK7 D97N knock-in cells. While 9 of the clones we characterised are single allele knockins, we did obtain one homozygous knock-in clone (22Rv1 CDK7 D97N CL3). Our results show that the 22Rv1 CDK7 D97N pool and the homozygous 22Rv1 CDK7 D97N CL3 are equally resistant to samuraciclib (Figure R1 here; not included in the manuscript). This supports our premise that a monoallelic dominant mutation is indeed sufficient to drive resistance to the non-covalent ATP-competitive CDK7 inhibitors.

This conclusion is also supported by the fact that the CDK4 D99N CRISPR knock-in single allele (MCF7-CDK4^{WT/D99N}) and complete knock-in (MCF7-CDK4^{D99N}) mutant cells show similar reduction in sensitivity to the CDK4/6i palbociclib (Figure 6D in the manuscript).

Figure R1: The top panel shows Sanger sequencing of the complete knock-in CDK7-D97N mutant clone (CL3). Arrowed and boxed is the G>A change resulting in the D97N mutation. Also arrowed are silent codon changes introduced into the CDK7 gene for PCR-directed identification of successful mutagenesis. The growth inhibition assay shown in the lower panel demonstrates a non-significant difference between response to Samuraciclib in the 22Rv1 CDK7-D97N pool and the CDK7-D97N complete knockin clone CL3.

IC₅₀: 22RV1 = 71.7 nM (5% CI: 62.6 – 82.3 nM), 22Rv1 CDK7-D99N CL3 = 1,415 nM (95% CI: 1,187 – 1,688 nM), 22Rv1 CDK7 D97N CRISPR pool = 1,530 nM (95% CI: 893.5 – 2578 nM). For comparison manuscript figure 1 provides the IC₅₀ values for 22RV1 = 107.1 nM (95% CI: 93.3 – 123.1 nM) and 22Rv1 SamR cells = 1,719 nM (95% CI: 1,473-2,005 nM).

2.) In Figure 5B, the authors show enrichment of the CDK12 (D819N) mutation in MCF7 cells following CRISPR knock-in and CR8 treatment. However, the chromatogram suggests the presence of additional mutations near this site. How can the authors conclude that the observed phenotype is solely due to D819N?

The reviewer is correct in noting the presence of additional mutations in the D819 CRISPR knock-in cells. We apologise for lack of clarity in our explanation of the knock-

in strategy which we have employed. The template oligo used for generating knock-in mutations was designed to include several silent codon changes. This allows us to perform knock-in specific PCR, facilitating our identification of successful knock-in pools and subsequently the isolation of knock-in clones. We used this strategy for all the knock-ins generated for this study. The additional (silent) mutations incorporated in the donor template are highlighted for the CDK7 D97N CRISPR (Figure 2A, Supplementary Figure 2A), CDK12 D819N (Supplementary Figure 4A) and CDK4 D99N (Supplementary Figure 5B). Agarose gels with products of the knock-in mutant-specific PCR for CDK12 D819N are shown in Supplementary Figure 4B and for CDK4 D99N in Supplementary Figure 5C. These agarose gel electrophoresis results show a PCR product following CRISPR editing but not in the parental MCF7 cells. The sequence for CDK7-D99N CL3 (Figure R1) also highlights these additional changes.

We have also added a statement further clarifying this point to the Materials and Methods section (p14). “Donor templates were designed to include non-coding substitutions to aid genome DNA PCR for identifying knockin-events and to facilitate cloning; these substitutions are noted in the relevant supplementary figures.”

3.) In Figure 1D, p-POLII levels in 22Rv1 cells do not show a clear reduction upon Samuraciclib treatment, despite the authors' claims. In contrast, Figure 2E demonstrates a more expected response in MCF7.

We have consistently observed weak inhibition of PolII phosphorylation by CDK7i in prostate cancer cell lines, necessitating higher concentrations of CDK7i, certainly compared to what we observe for breast cancer cell lines such as MCF7, as reported for prostate cancer cell lines (Constantin et al DOI: [10.1038/s41416-023-02252-8](https://doi.org/10.1038/s41416-023-02252-8)) and breast cancer cell lines (Patel et al DOI: [10.1158/1535-7163.MCT-16-0847](https://doi.org/10.1158/1535-7163.MCT-16-0847)). Nevertheless, quantification of immunoblotting of protein lysates from 3 independent experiments (Figure R2) clearly show that phosphorylation of the PolII-Ser5 phosphorylation (P-Ser5) is significantly reduced by 100 nM or 1000 nM samuraciclib in 22Rv1 cells but not in SamR cells. CDK2 phosphorylation is also significantly reduced in 22Rv1 cells but only with 1000 nM Samuraciclib in the SamR cells. In the case of both markers there is a significant reduction by SY1365 in SamR cells. These quantification data have now been added to the manuscript in supplementary Figure 1E.

Figure R2: Densitometric quantification of immunoblots from n=3 independent experiments in which 22Rv1 and 22Rv1-SamR cells were treated with the indicated

concentrations of Samuraciclib or SY1365. Signal intensities are shown relative to the appropriate vehicle (0 nM) controls.

4.) In Figure 6C, the authors do not specify whether CDK4 (D99N) CL1 cells are heterozygous or homozygous. They also claim a reduction in baseline Rb phosphorylation at S807/811 compared to WT, possibly due to altered CDK4 activity. However, no apparent difference in baseline phosphorylation is visible on the blot without Palbociclib treatment.

We apologise for the lack of clarity in Figure 6C. CL1 is homozygous and CL4 is heterozygous for the D99N mutation. We have amended the main text and figure legend to make this clearer and have listed the clone numbers in Figure 6C.

We did not intend to claim reduced baseline Rb1 phosphorylation at S807/811 in the mutant clones. To show RB1 phosphorylation, we now include the results of n=3 independent experiments, together with quantification, that clearly show that RB1 phosphorylation is significantly and dose-dependently reduced by palbociclib in MCF7 cells but that this inhibition is blunted in the D99N mutant cells, reductions in S807/811 phosphorylation not reaching significance, except at the highest dose. The quantification data have been included in a new supplementary figure 5E.

Figure R3: MCF7 and MCF7 CDK4D99N homozygous mutant clone CL1, were treated with palbociclib at the concentrations indicated, for 48 hours. Immunoblotting results for n =3 independent experiments are shown.

5.) Throughout the study, the impact of these mutations on kinase activity remains unclear. Assessing kinase activity in vitro or in homozygous clones would be necessary to clarify this. Although the cryo-EM structure suggests minimal functional impact, the authors still claim an effect on CDK4 activity without direct evidence.

The CDK4 D99N mutation does not affect growth of MCF7 cells, as indicated by the almost identical doubling times for the MCF7 cells and the homozygous D99N mutant clone (please see Fig. 6D and accompanying text). Thus, the D99N mutation does not impact CDK4 activity so drastically as to inhibit cell growth. We have included results for CDK4 activity in homozygous D99N mutant clones, together with quantification of n=3 independent experiments (as described for the reviewer's previous point), which clearly show that phosphorylation of the CDK4 substrate RB1 is not reduced in the homozygous (CL1) clone.

In the case of the CDK7 D97N mutation, we also don't see any impact of the mutation on growth, either for the 22Rv1 SamR cells described in Figure 1, or the CRISPR-generated homozygous CDK7^{D97N} mutant 22Rv1 clone (Figure R4). These results have been included in Supplementary Fig. 2 as parts E and F.

Figure R4: The indicated cells were assessed for growth over 150 hours. Doubling times were determined using nonlinear regression (curve fit) using Graphpad Prism for n=3 independent experiments. The doubling times and 95% confidence intervals are stated.

We have also carried out in vitro kinase assay data for CDK7 and CDK7-D97N CAK complexes used in the cryo-EM experiments. For this, we assessed the activities of the WT and mutant CAK complexes, the activity of CAK D97N is indeed lower than that of CAK WT in phosphorylation of CDK2. These results been added to Fig. 3 as parts D and E. Thus, the D97N mutation does indeed result in reduced CDK7 activity. We would hypothesise that the reduced kinase activity due to the mutation is sufficient for cell growth (as demonstrated by doubling times of the homozygous mutant clones) but is tolerated due to the selection pressure imparted by treatment with the CDK7i. Based on these findings, we also hypothesise that this mutation would result in active CDKs, but ones with reduced activities. The discussion has been amended to address these new findings.

Referee #2:

The paper by Lai et al., entitled "Mutation of CDK7 at a conserved residue identifies a

common mechanism of acquired resistance to CDDK inhibitors in cancer" describes the characterization of a spontaneous mutation in CDK7 that was discovered in a prostate cancer cell model of resistance to a CDK7 inhibitor, Samuraciclib, which has been shown by this team to have efficacy in ER+ breast cancer patients progressing on a prior CDK4/6 inhibitor treatment. Elegantly, this mutation was introduced in one breast cancer and two prostate cancer cell lines by targeted recombination, demonstrating that the mutation is sufficient for resistance. This residue is conserved in other CDKs and mutagenesis of the corresponding residue in CDK4, CDK6, and CDK12 also resulted in resistance to inhibitors of these enzymes. Mechanistically, although cryoEM structures of CDK7 did not reveal major changes in conformation of the protein but rather a greater structural heterogeneity at the active site, the author demonstrated using a nanoLuc assay that binding of all non-covalent ATP-competitive inhibitors occurred with lower affinity, explaining their lower potency in cell lines with the knock-in mutation.

The results of this detailed molecular study are clear and convincing. They are original and have important implications for resistance to a growing class of anticancer drugs, and thus should be of interest for a wide readership.

We thank the reviewer for their support of the findings we report, as well as their comments about the possible implications of our findings for resistance to CDK inhibitors.

My only reservation is that greater detail in the description of experimental approaches and results, as outlined below, would enhance the manuscript.

- Please provide some detail about the 22Rv1 model used as the resistance model.

The 22Rv1 cell line is a widely used prostate cancer model representing castrate-resistant prostate cancer (Sramkoski *et al*, 1999), where CDK7i could be explored as a potential treatment for this stage of the disease, representing patients with limited therapeutic options. The cell line is now described at its first mention at the start of the Results section.

- Description of Fig. 1D results with more granularity, perhaps including quantitation of bands in the Western analysis, would be desirable as some of the changes are more modest than those shown in Figure 2E.

We have expanded on the text describing the results in Figure 1D and have also included quantification of Western data from n=3 independent experiments (Supplementary Fig. 1E). Please see also response to reviewer 1, point 3

- Relative growth curves are provided but the assay used to monitor cell proliferation is not mentioned.

We apologise for this omission. A description of the development of SamR cells, as well as the cell proliferation assay details are now provided in Materials and Methods.

- Fig. 2E is not described

We can only apologise for this oversight and have now included text describing the results shown in Fig. 2E in the main text.

- While the CryoEM structure suggests that the mutation maintains kinase activity, please mention if the fitness of the cell lines bearing the mutant CDKs is affected.

We have assessed the fitness of the cell lines with new results in which we compared 22Rv1 and 22Rv1-SamR cells, with 22Rv1 cells with a CRISPR-generated CDK7-D97N mutant clone. The latter is homozygous for the D97N mutation. We show that all three cell lines have extremely similar doubling times (Supplementary Fig. 2E, F and accompanying text), which indicates that the mutation does not adversely affect cell growth *in vitro*. Similarly, the doubling times for MCF7, MCF7-CDK4^{WT/D99N} (heterozygous clone CL1) and MCF7-CDK4^{D99N} (homozygous mutant clone CL4) are 28.2, 28.5 and 28.4 hours, respectively. This suggests that the mutation does not impact the overall fitness of MCF7 cells.

- In the discussion, it would be worth comparing the clinical benefits of ATP-competitive vs covalent inhibitors, which are not affected by this mechanism of resistance.

Almost all CDKi that are currently in more advanced clinical trials or have been approved, are ATP-competitive inhibitors. This is likely, at least in part, to be due to the possible hepatic toxicity and immune responses that have historically been problematic for cancer drugs with reactive groups, as required for covalent inhibitors. Moreover, acquired resistance to covalent kinase inhibitors arising from mutation of the targeted residue is possible, exemplified by mutation of C481 commonly seen in CLL patients becoming resistant to the BTK inhibitor Ibrutinib. As proposed by the reviewer, we have included these points in the Discussion.

Referee #3:

The manuscript by Lai et al. reports the selection of specific mutations in the critical, conserved D97 Asp in CDK7 upon continuous treatment with ATP-competitors. Continuous treatment of prostate cancer cells with Samuraciclib, a non-covalent ATP-competitive CDK7i, leads to the selection of resistant clones characterized by a single allele alteration in exon 5 (c.289G>A; p.Asp97Asn; D97N). This change results in resistance to multiple non-covalent, ATP-competitive CDK7i, but not to covalent inhibitors. The CDK7-D97N mutation does not induce major alterations of the CDK7

active site, consistent with the observation that the mutant enzyme might be still active. However, the affinity of the non-covalent inhibitors to CDK7-D97N is reduced. These data are reproduced in different cell types after CRISPR-mediated gene editing of this residue in CDK7.

Since the Asp residue is conserved across the CDK family, the authors repeat CRISPR-mediated gene editing of CDK12, CDK4 or CDK6 to demonstrate that a similar change to Asn reduces the binding and effect of non-covalent ATP-competitors. Similarly, the change per se does not seem to significantly alter the growth of the untreated cells, suggesting active kinases in the presence of that mutation. Overall, these data suggest that the Asp to Asn mutation in the critical Asp of the hinge domain of CDKs may have mild effects on the activity of the kinase, while decreasing the affinity for specific ATP-competitors.

The data are interesting, and the fact that specific conservative mutations in this residue may alter the binding of ATP or ATP-competitors is not unexpected. Yet, The description of Asp-to-Asn mutations that maintain the endogenous function of the protein while decreasing the response to clinically-relevant inhibitors open the possibility of finding similar mutations in patients. Since these mutations have not been found in patients yet, it remains to be seen whether this is a reality. There are a couple of unexplored reasons why this is not frequent (or even seen) in patients even after whole-exome sequencing (in breast cancer; Wander et al., 2020 among others).

If I understand properly the data, the manuscript describes the selection of this mutation in a single clone (or pool?), 22Rv1-SamR, in Figure 1. The rest of assays are performed after CRISPR-mediated gene editing. How many times that mutation has been observed in different clones? And in different cell lines? Could it be possible that this mutation is pre-existing in 22Rv1 cells? Could it be detected in untreated 22Rv1 prostate cancer using high-depth sequencing methods?

We thank the reviewer for their interest in our findings. We undertook the selection in 22Rv1 and in LNCaP cells. The mutation emerged in 22Rv1 cells but was not observed in LNCaP cells. It was for this reason that we undertook the experiments to introduce the CDK7 mutation in several cell lines, so that we could confirm that this change was necessary and sufficient for resistance to CDK7i. Our RNA-seq analysis of 22Rv1 identified 936 sequence reads in the region of the CDK7 gene encoding c.289G. Not one read had an alteration in this sequence, which implies that the c.289G>A alteration is indeed rare and if pre-existing, that it is present at a frequency of $\leq 0.1\%$.

The central Asp in the CDK hinge is a critical residue to CDK activity. Although mutant clones seem to grow similarly in the absence of inhibitors, a more direct comparison on the relative binding of ATP versus different ATP-competitors would be informative. In addition, a direct quantification of phosphorylation activity using more direct molecular quantification methods (rather than WB in cells) to compare ATP versus ATP-competitor activity would also be important to understand whether these mutations are never seen in vivo due to minor but significant defects in the endogenous kinase

activity.

As the reviewer points out, mutant cells, including homozygous mutant clones grow similarly to the WT cells, findings that we have more clearly shown in the revised manuscript (please see responses to queries from reviewers 1 and 2). For CDK7 and CDK4, we have used in-cell kinase assays to demonstrate reduced binding of ATP-competitive inhibitors (Fig. 3A-D). To address the central point of the reviewer's comment as to whether the mutant CDKs are less active than the WT CDK, we have undertaken an *in vitro* kinase assay for CAK-WT and CAK-D97N (shown in new Fig. 3E-F). These results show that the D97N mutation does indeed reduce kinase activity. We would, however, note that the mutant CDK7 and CDK4 are, nonetheless sufficiently active that there is no difference in growth of homozygous mutant CDK7 and CDK4 clones.

Related to the previous comment, the relative activity is hard to compare in WB studies as some CDK7i do not seem to be effective on RNAPol CTD sites even in parental cells (see Fig. 1D). A quantification of these signals in several repetitions would also help here.

As suggested by the reviewer, we have carried out several independent experiments, to perform quantification of the WB. As shown in Fig. R2 (above), immunoblotting of protein lysates from 3 independent experiments show that phosphorylation of the PolII-Ser5 phosphorylation (P-Ser5) is significantly reduced by 100 nM or 1000 nM samuraciclib in 22Rv1 cells but not in SamR cells. CDK2 phosphorylation is also significantly reduced in 22Rv1 cells but only with 1000 nM Samuraciclib in the SamR cells. In the case of both markers there is a significant reduction by SY1365 in SamR cells. These quantification data have now been added to the manuscript in supplementary Figure 1E.

The fact that these mutations have not been seen in patients (perhaps due to reasons discussed in the previous points) deserves some discussion. The Discussion section is very short in the present document (apart from discussing the Persky et al., 2020 study), and it would benefit from reduced sentences dedicated to repeat the results in favor of a more clinically discussion: Why these discussions have not been seen in patients? Would similar mutations affect differentially different drugs? Could eventually these mutations be targeted with complementary drugs that do not require similar binding to the same residues? Would other mutations in Asp be lethal (as this residue is typically mutated to generate kinase-dead CDKs)?

We have refocussed the Discussion to include the points raised by referee 2, as well as discussing the clinical implications, as suggested by the reviewer.

We would note that CDK4/6i have been widely used in estrogen receptor positive (ER+) breast cancer, with many studies reporting on mutational analysis, including the CDK4 and CDK6 genes, with no reports of CDK4 or CDK6 mutations. However, ER+ breast cancer patients are treated with CDK4/6i in combination with hormone therapies targeting ER and other resistance mechanisms have been reported for CDK4/6i.

While we recognise that the mutations we describe have not yet been identified in cancers, this is likely to reflect the fact that very few patients have so far been treated with CDK7i. CDK12i and CDK2i have only recently started in clinical trials so it is too early for resistance mechanisms to be evident. Notwithstanding, what our work demonstrates is the clear potential for a resistance mechanism in which resistance to CDK inhibitors arises due to dominant mutations in the targeted CDK gene. We would also emphasise the point that biochemical studies often presage clinical discoveries. Another area we are working on is ER, exemplifies this point. Biochemical studies identified amino acid substitutions that would be capable of causing resistance to hormone therapies (Weis 1996 Nov;10(11):1388-98. doi: 10.1210/mend.10.11.8923465) many years prior to the discovery of the very same mutations in ER+ metastatic breast cancer patients (Li et al Cell Rep. 2013 Sep 26;4(6):1116-30. doi: 10.1016/j.celrep.2013.08.022; Robinson et al 2013 DOI: [10.1038/ng.2823](https://doi.org/10.1038/ng.2823); Toy et al 2013 DOI: [10.1038/ng.2822](https://doi.org/10.1038/ng.2822)).

Could perhaps the title reflect that data were generated in vitro? The "common" word is also misleading here. It may apply to other CDKs but is not commonly found either in vitro or in vivo.. In general, I acknowledge the value of the present study but, in the present form, it remains closer to a biochemical curiosity than a relevant description of current mechanisms of resistance in cancer.

We have changed the title to reflect the point made by the reviewer. The new title is "Resistance to CDK7 inhibitors directed by acquired mutation of a conserved residue in cancer cells". A more explicit title was not possible due to character limit of 100.

Dear Dr Ali,

Thank you for submitting your revised manuscript (EMBOJ-2024-119953R) to The EMBO Journal, as well for your patience with our feedback. Your amended study was sent back to the three referees for their scientific reassessment, and we have received re-reports from all of them, which I enclose below. As you will see, the reviewers state that the work has been substantially enhanced by the revisions and they are now broadly in favour of publication, pending minor amendments.

Thus, we are pleased to inform you that your manuscript has been accepted in principle for publication in The EMBO Journal.

Please carefully consider the remaining minor points raised by referee #1 by adjusting the discussion of the findings where appropriate.

Also, we now need you to take care of a number of issues related to formatting and data presentation as detailed below, which should be addressed at re-submission.

Please contact me at any time if you have additional questions related to below points.

As you might have seen on our web page, every paper at the EMBO Journal now includes a 'Synopsis', displayed on the html and freely accessible to all readers. The synopsis includes a 'model' figure as well as 2-5 one-short-sentence bullet points that summarize the article. I would appreciate if you could provide this figure and the bullet points.

Thank you for giving us the chance to consider your manuscript for The EMBO Journal. I look forward to your final revision.

Again, please contact me at any time if you need any help or have further questions.

Best regards,

Daniel Klimmeck

>> Please add up to five keywords to your study.

>> Limit the abstract to maximally 175 words.

>> Author Contributions: Remove the author contributions information from the manuscript text. Note that CRediT has replaced the traditional author contributions section as of now because it offers a systematic machine-readable author contributions format that allows for more effective research assessment. and use the free text boxes beneath each contributing author's name to add specific details on the author's contribution.

More information is available in our guide to authors.

>> Please rename the current 'Competing Interests' section to 'Disclosure and Competing Interests Statement' and place it after the Acknowledgements.

>> 'Material and Methods' should be renamed to 'Methods'.

>> Data availability section: please adjust the title of the current 'Data and code availability' section to 'Data availability'.

>> The manuscript sections should be in the following order: Title page - Abstract & Keywords - Introduction - Results - Discussion - Methods - Data Availability - Acknowledgments - Disclosure Statement & Competing Interests - References - Figure Legends - (Main Tables with legends if applicable) - Expanded View Figure Legends.

>> Figure callouts: Please ensure that the figures and panels are called out in sequential order. Currently, there is a callout for the missing Supplementary Table 1; figure panels for EV figures not called out; callouts and figure legends should be renamed to Figure EV1-EV5 instead of EV figure 1-5.

>> Add a separate 'Statistical Analysis' section to the Methods part, detailing the algorithms and statistical tests applied.

>> Add a Reagents and Tools table to the Methods section, as a separate file using the existing template in the Guide For Authors, listing key reagents, experimental models, software and relevant equipment.

>> Please provide a completed source data checklist for the study as to the separate request e-mail by our office team.

>> Consider additional changes and comments from our production team as indicated below:

- Figure Legends (main + EV): 1. Please define the annotated p values ****/**/*/* as well as provide the exact p-values for the same in the legend of figure EV1 E, EV5 E as appropriate.

2. Please note that the exact p values are not provided in the legend of figure 3F

3. Please indicate the statistical test used for data analysis in the legends of figures EV1 E, EV5 E.

4. Please note that scale bar and its definition are missing for figure EV5 F.

Referee #1:

The authors have addressed all of my concerns, and the manuscript has improved substantially. I would have appreciated the inclusion of an in vitro CDK4 activity assay, as it would have further strengthened the data. However, I find the doubling time measurements to be sufficient. The authors should note that CDK6 can compensate for CDK4 function, which may explain why the CDK4 D99N cells are still proliferating. I suggest that the authors consider mentioning this point in the discussion.

Referee #2:

All my comments have been satisfactorily addressed.

Referee #3:

The authors have improved the manuscript by adding additional controls and evidences. Specifically, I believe that characterizing the relative impact of these mutations on the kinase activity was important and the authors have clarified this point in detail. I still believe that the fact that these mutations have only been found in vitro (and are quite infrequent) limit a bit the impact of the manuscript. There are a few hundred of breast cancer patients treated with CDK4/6i sequenced (panel, WES or WGS) and these mutations have not been found in CDK4 or CDK6. As the authors indicate, it is to early to know in other CDKs.

Dear Dr Klimmeck,

On behalf of myself and my co-authors, I would like to state that we are delighted that our manuscript has been accepted for publication. We have looked over the reviewers' comments and the formatting changes requested, and I address all points below.

Please carefully consider the remaining minor points raised by referee #1 by adjusting the discussion of the findings where appropriate.

Referee #1:

The authors have addressed all of my concerns, and the manuscript has improved substantially. I would have appreciated the inclusion of an in vitro CDK4 activity assay, as it would have further strengthened the data. However, I find the doubling time measurements to be sufficient. The authors should note that CDK6 can compensate for CDK4 function, which may explain why the CDK4 D99N cells are still proliferating. I suggest that the authors consider mentioning this point in the discussion.

We are grateful to the reviewer for their comment and have added the following text in the Discussion (p10 of revised manuscript), which complies with their request. "In the case of CDK4, another explanation for the lack of a proliferation block in mutant cells could be CDK6, since CDK4 and CDK6 have apparent redundant roles in G1 progression, at least in breast cancer cells. However, our analysis showed that CDK6 expression is low/absent in MCF7 cells, which were used for generating the CDK4 mutation."

Dear Dr Ali,

Thank you for submitting the revised version of your manuscript. I have now evaluated your amended manuscript and concluded that the remaining minor concerns have been sufficiently addressed.

I am thus pleased to inform you that your manuscript has been accepted for publication in the EMBO Journal.

On a different note, I would like to alert you that EMBO Press offers a format for a video-synopsis of work published with us, which essentially is a short, author-generated film explaining the core findings in hand drawings, and, as we believe, can be very useful to increase visibility of the work. Please see the following link for representative examples and their integration into the article web page:

<https://www.embopress.org/doi/full/10.15252/emj.2019103932>

Best regards,

Daniel Klimmeck

Daniel Klimmeck, PhD
Senior Editor
The EMBO Journal
EMBO
Postfach 1022-40
Meyerhofstrasse 1
D-69117 Heidelberg
contact@embojournal.org
